# Fair Matroid Selection

**Kiarash Banihashem**
Department of Computer Science
University of Maryland
kiarash@umd.edu

**MohammadTaghi Hajiaghayi**
Department of Computer Science
University of Maryland
hajiaghayi@gmail.com

**Danny Mittal**
Department of Computer Science
University of Maryland
dannymittal@gmail.com

## Abstract

We investigate the problem of sequentially selecting elements of an unknown matroid in an online manner to form an independent set, with the goal of maximizing the minimum probability of acceptance across all elements, a property we define as $f$-fairness. Under adversarial arrival orders, we design an $\alpha(\ln k + 1)$-fair algorithm, where $\alpha$ is the arboricity of the matroid and $k$ is the rank, a result that is nearly optimal. For laminar matroids, we develop a $(2\alpha - 1)$-fair algorithm, which is optimal up to constant factors, achieved through a novel online coloring scheme. In the random arrival order setting, we achieve a $(4 + o(1))\alpha$-fair algorithm for graphic matroids, matching the optimal result up to constant factors, relying on a novel technique for learning a degeneracy ordering using a sampled subset of edges. We further generalize our result to $p$-matchoids, obtaining a $\beta(p \ln k + 1)$-fair algorithm for the adversarial arrival model, where $\beta$ is the optimal offline fairness. Notably, all our results can be extended to a setting with no prior knowledge of the matroid with only a logarithmic increase in the fairness factor.

## 1 Introduction

One of the most central problems in online decision making is the *secretary problem* introduced in the seminal result of Dynkin [Dyn63]. In this problem, a sequence of values arrives online in a random order, and the goal is to select the maximum value. It is well known that the optimal strategy selects the maximum value with probability $1/e$. A natural generalization allows selecting multiple elements, subject to combinatorial constraints, and competing with the offline optimum in expectation. One of the central open questions in this area is the matroid secretary conjecture of Babaioff et al. [BIK07, BIKK18], which posits that a constant-competitive algorithm exists when the feasibility constraint is given by a matroid [1]. Despite extensive research, this conjecture remains unresolved. However, an important limitation of the standard matroid secretary problem is that it solely maximizes the sum of accepted elements, potentially leading to highly unequal outcomes—an issue of particular concern from a fairness perspective.

Motivated by this, we introduce and study the *fair matroid selection* problem. In this problem, elements from an unknown matroid arrive online, and an algorithm must select elements irrevocably while respecting the matroid's independence constraints. Said algorithm may learn the matroid's structure online via independence queries on sets containing only elements which have already

---

[1] A matroid is a tuple $M = (E, \mathcal{I})$, where $E$ is a finite ground set and $\mathcal{I} \subseteq 2^E$ satisfies: (1) If $I \in \mathcal{I}$ and $I' \subseteq I$, then $I' \in \mathcal{I}$; (2) If $I, J \in \mathcal{I}$ with $|I| < |J|$, then there exists $e \in J \setminus I$ such that $I \cup \{e\} \in \mathcal{I}$.

39th Conference on Neural Information Processing Systems (NeurIPS 2025).

arrived. We define an algorithm alg as $f$-fair (where $f > 1$) if $\min_e \Pr\left[e \in \text{alg}\right] \geq f^{-1}$. That is, every element has a guaranteed selection probability of at least $f^{-1}$. We explore this problem under both adversarial and random arrival models, investigating the minimum achievable fairness parameter $f$ in each case.

More broadly, online selection problems have been widely studied under different assumptions for both random and adversarial arrival models [Din13, CFH$^+$19, GS20]. The general framework involves a set of elements from a combinatorial structure arriving sequentially, each associated with a weight. The goal is to irrevocably select a feasible subset of elements to maximize the total weight. A common application of these problems arises in scenarios where elements represent agents and weights correspond to agent utilities. From this perspective, maximizing total weight translates naturally to maximizing social welfare, i.e., the sum of all agents' utilities.

While social welfare is a natural objective, it does not account for how utility is distributed among agents. In many settings, a purely welfare-maximizing allocation can result in extreme disparities, where some agents receive little to no utility. This motivates the study of fairness-aware algorithms, which aim to ensure that every agent receives a meaningful share of the available utility. Fair allocation problems have been studied for decades (e.g., see Varian [Var74]) and appear in many real-world domains, including job assignments, online advertising auctions, and resource-limited procurement, where fairness constraints help prevent highly uneven outcomes [PV22, LWZ23, FJU24, CEEV24].

A widely studied notion of fairness is *max-min fairness*, which aims to maximize the minimum utility received by any agent. In the context of offline fair allocation, this problem is generally referred to as the Santa Claus problem [BS06], though variants had been previously explored under different names [LMMS04, BD05], where the goal is to distribute a set of items among agents while ensuring that the least satisfied agent receives as much utility as possible. Multiple works have examined this problem from a computational perspective, focusing on designing efficient approximation algorithms (see Section A for further discussion). Recent works have studied the Santa Claus problem in online settings as well [SHPK22]. However, the problem formulations in these works differ significantly from the secretary problem and the closely related prophet inequalities problem. In most of these works, agents are not selected in a traditional sense; instead, each agent is allocated a set of items.

Given the significance of the matroid secretary problem, numerous works have sought to develop constant-competitive algorithms for specific subclasses of matroids. Two particularly well-studied cases are laminar matroids and graphic matroids, both of which have been the focus of extensive research, with many works progressively improving the competitive ratio [BIK07, BDG$^+$09, KP09, IW11, JSZ13, MTW16, STV21, HPZ23, BLSV24, BHK$^+$25]. In this work, we take an analogous approach, focusing on special cases—but in the context of fair matroid selection—where we obtain improved results.

## 1.1 Our contributions

We first show that for general matroids, the optimal fairness can be nearly tightly characterized in terms of the *arboricity* of the matroid, defined as the minimum number of independent sets required to cover the matroid, which we denote with $\alpha$. The idea behind this result is to exploit Edmonds' formula (Theorem 6) for $\alpha$, which makes a natural lower bound for $\alpha$ tight. The below theorem is restated in the body as Theorem 7.

**Theorem 1.** *An $f$-fair algorithm for fair matroid selection must satisfy $f > \alpha - 1$ even if it is allowed to know all elements of the matroid upfront. Furthermore, if all elements are known upfront, there exists a natural $\alpha$-fair algorithm.*

We then move to our central result, showing that we can in fact be competitive with an offline algorithm even in a fully adversarial online setting, losing only a factor of $O(\log k)$ in fairness.

**Theorem 2.** *There exists an $\lceil \alpha(\ln k + 1) \rceil$-fair algorithm for the adversarial order fair matroid selection problem when $\alpha$ is known upfront. Furthermore, there exists an $\tilde{O}(\alpha \log k)$-fair algorithm that knows absolutely nothing upfront.*

This is proven as Theorem 10 and Corollary 2. There are two key ideas behind the associated algorithm. The first is to reduce fair matroid selection to an online coloring problem, where we must color arriving elements so that each color forms an independent set: we can then pick a color uniformly at random in advance of the elements' arrival, and accept all elements with that color. We

then color using a simple greedy algorithm; the second idea comes in the analysis of this greedy algorithm, where the fact that the first color forms a maximal independent set leads us to argue that each color removes $\frac{1}{\alpha}$ of the remaining elements. [2]

We then consider the special case of *laminar matroids* which are characterized by a laminar family of sets over the ground set, each with an associated budget, so that an independent set in the matroid does not have more elements from any set in the laminar family than the budget allows for. We study the *fair laminar matroid selection* problem where the elements arrive online, each revealing the sets containing it. Our main result is the following theorem which resolves the problem in this special case up to constant factors.

**Theorem 3.** *There exists a $2\alpha$-fair algorithm for the adversarial order fair laminar matroid selection problem when $\alpha$ is known in advance. Furthermore, there exists a $2h(\alpha)$-fair algorithm with absolutely no upfront knowledge, where $h(\alpha)$ can be chosen to be the fairness of any algorithm for fair matroid selection on rank-1 matroids; in particular, $h(\alpha)$ can be $\tilde{O}(\alpha)$.*

This is proven as Theorem 11 and Corollary 3. This algorithm once again makes use of the reduction to the coloring problem introduced for the general algorithm. We then exploit the tree-like structure of laminar matroids to simulate an Euler tour ordering of the elements of the laminar matroid online, coloring the elements so that any $\alpha$ consecutive elements have distinct colors, which then guarantees independence for each color's set of elements.

We next turn our attention to the random arrival order model and obtain nearly tight results for the *fair graphic matroid selection* problem. In this problem the elements of the matroid are edges of a graph and a set is independent if and only if it contains no cycles. The edges arrive online, revealing their endpoints. Our main result is the following theorem.

**Theorem 4.** *There exists a $(4 + o(1))\alpha$-fair algorithm for the random order fair matroid selection on graphic matroids when $\alpha$ is known in advance. Furthermore, there exists an $O(\alpha)$-fair algorithm that does not know $\alpha$ in advance.*

This is proven as Theorem 15 and Corollary 4. This result is our most technically involved. We first note that $\alpha$ can be 2-approximated by the *degeneracy* of a graph, i.e. the minimum integer $d$ such that the vertices can be ordered to have at most $d$ edges to preceding vertices; such an ordering, known in advance, would immediately imply a $d$-fair algorithm. We then carefully learn an approximate such ordering online based on a sample of the edges, showing how the specific manner in which we construct the ordering guarantees that each vertex, with high probability, has at most $d + o(d)$ edges to preceding vertices.

We finally generalize our result for general matroids for the *fair matchoid selection problem*, where the underlying constraint is a matchoid instead of a matroid. Matchoids are a well-studied structure for combining multiple matroid constraints; briefly, a $p$-matchoid imposes multiple matroid constraints on a single ground set, such that any element of the ground set is constrained by at most $p$ matroids. We prove the following theorem.

**Theorem 5.** *For any $p \geq 1$, there exists a $\lceil \beta(p \ln k + 1) \rceil$-fair algorithm for the adversarial order fair matchoid selection problem over $p$-matchoids with rank $k$ where $\beta$ is the optimal offline fairness, when $p, k, \beta$ are known in advance. Furthermore, there exists an $\tilde{O}(\beta p \log k)$-fair algorithm that knows absolutely nothing in advance.*

This is proven as Theorem 17 and Corollary 5 in Appendix B. This result relies on many of the same ideas as the result for the single-matroid case; the most notable difference comes from the fact that a maximal set in a $p$-matchoid only $p$-approximates a maximum set in a $p$-matchoid, roughly explaining why we lose an additional factor of $p$ in fairness.

**Map of the paper.** The remainder of the paper is organized as follows. In Section 2, we relate our notion of fairness to the concept of *arboricity* of graphs and matroids. In Section 3, we study the adversarial order model. In Section 4, we study the random order model. Due to space constraints, we defer parts of the proofs, as well as a discussion of further related work, to the Appendix.

---

[2]We note that the greedy approach has also been considered by prior work on online matroid coloring [FKT89].

## 2 Characterizing Fairness

As described in the introduction, we characterize algorithms for fair matroid selection, regardless of the variant, by their *fairness*; specifically, we say that an algorithm for a particular variant of fair matroid selection is $f$-fair if it guarantees that any element of the matroid to be presented is selected with probability at least $\frac{1}{f}$. Thus, we aim to minimize the value of $f$, both asymptotically and by constant factors.

A natural first question to ask is what kind of limits exist on $f$: clearly, no algorithm can be better than 1-fair, as an element cannot be selected with probability higher than 1. Even further than this however, if we consider a rank-1 matroid with $n$ elements, as only 1 element can be selected, some element must be selected with probability at most $\frac{1}{n}$, and so no algorithm can be better than $n$-fair, even if we discard the online setting and allow the algorithm to see all elements in advance. We thus seek a way of characterizing what an "optimal" value of $f$ looks like as a function of the matroid to be presented, and the quantity that arises to serve this purpose is the *arboricity* of the matroid.

The *arboricity*, which we will refer to as $\alpha$, is traditionally defined on a graph $G$ to be the minimum number of sets into which we must partition the edges of $G$ so that each set is a forest, i.e. is free of cycles. By analogy with graphic matroids, the arboricity can be generalized to matroids as follows:

**Definition 1.** *For any natural number $\alpha$, we say that a matroid $M = (E, \mathcal{I})$ is $\alpha$-arboric if there exists a partition $E = A_1 \sqcup \cdots \sqcup A_\alpha$ of its elements such that for all $j = 1, \ldots, \alpha$, $A_j$ is independent. The* arboricity *of $M$ is the minimum $\alpha$ such that $M$ is $\alpha$-arboric.*

The above quantity is not usually referred to as the arboricity, and indeed does not seem to have a commonly used name. In spite of this, it is well studied, most notably by Edmonds [Edm65], who in 1965 simultaneously proved the following formula for the arboricity and gave an algorithm for computing the corresponding partition.

**Theorem 6** (Edmonds)**.** *For any matroid $M = (E, \mathcal{I})$, let $\alpha$ be its arboricity. Then,*

$$\alpha = \left\lceil \max_{A \subseteq E} \frac{|A|}{\operatorname{rank} A} \right\rceil;$$

*that is, $\alpha$ is the smallest integer which is at least the ratio of the size of $A$ to the rank of $A$ for all sets $A$ of elements of the matroid.*

Theorem 6 is fundamental for fair matroid selection as it can be used to show that the optimal fairness of an offline algorithm, i.e. an algorithm which is allowed to view the entire matroid upfront before making any decisions, is within an additive factor of 1 of the arboricity of the matroid:

**Theorem 7.** *For any matroid $M = (E, \mathcal{I})$, let $\alpha$ be its arboricity. Then there exists an $\alpha$-fair offline algorithm for fair matroid selection on $M$, while there does not exist an $(\alpha - 1)$-fair offline algorithm for fair matroid selection on $M$. Thus, the optimal fairness of an offline algorithm for fair matroid selection on $M$ lies in the range $(\alpha - 1, \alpha]$.*

The idea behind this theorem is that an offline algorithm can simply partition the matroid into $\alpha$ independent sets (we remark that this can be done efficiently using Edmonds' algorithm), then pick one of these sets uniformly at random; meanwhile, Theorem 6 tells us that a set $S$ of elements exists such that the number of elements in $S$ is nearly $\alpha$ times as much as the number of elements of $S$ that can be taken at once, giving the lower bound. A formal proof is located in Appendix C.1.

The arboricity $\alpha$ is therefore a natural parameter by which to characterize the fairness of algorithms for fair matroid selection – an $\alpha$-fair algorithm can be said to be $\frac{\alpha}{\alpha-1} = 1 + o(1)$-competitive with the optimal offline algorithm for fair matroid selection, and for any $\alpha$ there exist matroids (even more simply, rank-1 matroids) for which $\alpha$-fairness is tight. Therefore, a natural goal for algorithms for fair matroid selection is to achieve or at least approach $\alpha$-fairness.

## 3 Adversarial Order Fair Matroid Selection

We now move to the main problem we consider, which is the adversarial order variant of fair matroid selection. The formal description of the problem is as follows: a matroid $M = (E, \mathcal{I})$ is chosen and $E$ is ordered, unknown to the algorithm. The algorithm is presented with the elements of $E$ in

said order, which the algorithm must accept or deny online, subject to the constraint that the set of elements it accepts must be independent. At any point in time, the algorithm can access the matroid via independence queries containing only elements which have already been revealed. In the most natural formulation of this problem, the algorithm is *totally oblivious*, meaning that it receives no information upfront about the matroid it will observe – it does not even know the number of elements in the matroid. As we will see, even this totally oblivious variant of fair matroid selection is tractable. However, when designing algorithms, it does turn out to be useful to have knowledge of certain properties of the matroid upfront. The most commonly useful will be the arboricity $\alpha$, being that it characterizes the optimal fairness achievable; in addition, it is sometimes useful to know the rank $k$ of the matroid. We would therefore like to design algorithms in a setting in which such properties are known – such algorithms will be described, for example, as $\alpha$-knowing – and then somehow adapt them to the more difficult setting in which nothing is known upfront.

We show that such an adaptation is possible, and furthermore characterize how said characterization can be performed optimally. We define a class of *feasible* functions; feasible functions will determine how much we lose in fairness when adapting an algorithm that knows some parameter upfront to be totally oblivious.

**Definition 2.** *A function $h : \mathbb{N}_+ \to \mathbb{R}$ is said to be* feasible *if $h$ is nondecreasing and $\sum_{j=1}^{\infty} \frac{1}{h(j)} \leq 1$.*

More strongly, in adapting an algorithm aware of some parameter $p$ to one that is oblivious, we can lose a factor of at most $f(p)$ if and only if $f$ is feasible; this is expressed in the following theorems. Theorem 8 shows how adaptation can be performed for any algorithm with a loss of only $f$ if $f$ is feasible, while Theorem 9 shows how a loss of some feasible $f$ is necessary even for the simple case of rank-1 matroids. Proofs of both theorems are located in Appendix C.2.

**Theorem 8.** *Suppose that for some class $C$ of matroids, we have a function $f : C \to \mathbb{Z}_+$ defining a property of matroids in $C$. Further suppose that we are given an algorithm $A_l$ parameterized by a positive integer $l$ and a constant $c \in \mathbb{R}_+$ such that the algorithm $A_l$ is $cl$-fair when the input matroid $M \in C$ satisfies $f(M) \leq l$. Then for any feasible $h$, there exists a totally oblivious fair matroid selection algorithm $A'$ which is $c \cdot h(f(M))$-fair on $M \in C$.*

**Theorem 9.** *Let $A$ be a totally oblivious fair matroid selection algorithm for the class of rank-1 matroids, such that $A$ is $h(\alpha)$-competitive when the matroid is $\alpha$-arboric. Then $h$ is feasible.*

Concretely, Theorem 8 in fact allows us to make algorithms totally oblivious while only losing logarithmic terms in the fairness, which we express in the below corollary (also proven in Appendix C.2):

**Corollary 1.** *Suppose that for some class $C$ of matroids, we have a function $f : C \to \mathbb{Z}_+$ defining a property of matroids in $C$. Further suppose that we are given an algorithm $A_k$ parameterized by a positive integer $k$ and a constant $c \in \mathbb{R}_+$ such that the algorithm $A_k$ is $ck$-fair when the input matroid $M \in C$ satisfies $f(M) \leq k$. Then there exists a totally oblivious fair matroid selection algorithm $A'$ which is $O(f(M) \cdot \log^2 f(M)) = \tilde{O}(f(M))$-fair on $M \in C$.*

Having shown how algorithms can be adapted to be totally oblivious in an optimal way, we now focus our attention on the setting where algorithms have upfront knowledge of properties of the matroid. The remainder of the section is organized as follows: subsection 3.1 deals with our main result for general matroids, while the following subsections deal with algorithms for specific classes of matroids – subsection 3.2 presents a near optimal algorithm for the class of laminar matroids. We also note that for the further special case of uniform matroids, there exists an $\alpha$-knowing, $\alpha$-fair algorithm, which then both is optimal for an $\alpha$-knowing algorithm and implies an optimal totally oblivious algorithm by Theorem 8; a proof of this can be found in Appendix C.10.

## 3.1 General matroids

The main result of this paper is the below algorithm for adversarial order fair matroid selection on general matroids. This algorithm shows that even the general adversarial order fair matroid selection problem is tractable, which is expressed in the below theorem.

**Theorem 10.** *Let $\alpha, k$ represent the arboricity and rank of a matroid respectively. Then Algorithm 1 is an $\lceil \alpha(\ln k + 1) \rceil$-knowing algorithm for adversarial order fair matroid selection that is $\lceil \alpha(\ln k + 1) \rceil$-fair.*

We first note that we can apply Theorem 8 to obtain an oblivious algorithm for fair matroid selection:

**Algorithm 1:** General algorithm for adversarial order fair matroid selection.

---

Let $m = \lceil \alpha(\ln k + 1) \rceil$.
For each $j = 1, \ldots, m$, initialize the set $A_j$ as empty.
Choose $r$ uniformly at random from $1, \ldots, m$.
**for** *e being presented* **do**
    Let $j$ be the minimum among $1, \ldots, m$ such that $A_j \cup \{e\}$ is independent.
    Insert $e$ into $A_j$.
    **if** $j = r$ **then**
        Accept $e$.

---

**Corollary 2.** *There exists a totally oblivious algorithm for adversarial order fair matroid selection which is $\tilde{O}(\alpha \log k)$-fair.*

Thus, there exists an algorithm for the most pessimistic variant of fair matroid selection, assuming no upfront knowledge of the matroid and an adversarial ordering, which only loses in fairness compared to the optimal $\alpha$ by logarithmic factors in $\alpha$ and $k$.

We now describe the two key ideas underlying Algorithm 1 and its analysis. The first idea is to convert the fair matroid selection problem into an online coloring problem: we aim to color all arriving elements using at most $\alpha(\ln k + 1)$ colors such that each color class forms an independent set. Then, by choosing a color uniformly at random and selecting all elements of that color, we ensure fairness (since every element is colored) and validity (since each color class is independent). Crucially, the color can be chosen in advance, before any elements are revealed. The second idea shows how this coloring can be done greedily. The analysis begins with observing that the first set $A_1$ must be a maximal independent set, and thus have rank $k$. Since the matroid is $\alpha$-arboric, it can be partitioned into $\alpha$ independent sets, each of size at most $k$, implying that $M$ has at most $\alpha k$ elements and so $A_1$ covers at least a $\frac{1}{\alpha}$ fraction of them. By repeatedly applying this argument, each iteration removes at least a $\frac{1}{\alpha}$ fraction of the remaining elements, and thus the number of iterations needed to cover all elements is at most $\ln k + 1$.

The full proofs of Theorem 10 and Corollary 2 are located in Appendix C.3.

### 3.2 Laminar matroids

The *laminar matroids* are the class of matroids that can be defined as follows. Let $M = (E, \mathcal{I})$ be the matroid in question. Then there exists an integer $m$ and two sequences $S_1, \ldots, S_m$ and $r_1, \ldots, r_m$ such that for each $j = 1, \ldots, m$, $S_j \subseteq E$ and $r_j$ is a positive integer. We further constrain that the family of sets $F = \{S_1, \ldots, S_m\}$ is a *laminar* family, meaning that for any two $S, T \in F$, if $S$ and $T$ have nonempty intersection, then either $S \subseteq T$ or $T \subseteq S$. We then say that a set $A \subseteq E$ is independent iff for all $j = 1, \ldots, m$, $|A \cap S_j| \le r_j$.

In this way, a laminar matroid represents a hierarchy of restrictions – we can build a hierarchy (alternatively, a rooted tree) with each element of the matroid being at the bottom of the hierarchy (i.e. a leaf), where each level of the hierarchy imposes a constraint on the amount of elements within that level that can be in an independent set (i.e. each vertex imposes a constraint on the amount of its descendants that an independent set can contain). Laminar matroids thus provide a rich structure to work with.

There, in fact, exists a very practical motivation for considering fair matroid selection under the hierarchy of restrictions imposed by a laminar matroid. Consider a public magnet school drawing its applicants from multiple counties in a metropolitan area. In order to address concerns about geographical diversity, the school employs a "geographical lottery" system of admissions. Applicants are first deemed acceptable based on academic criteria. Acceptable applicants are then selected at random, subject to constraints: a cap on the total number of applicants to be accepted, a cap on the number of applicants to be accepted from each county based on the funding they can provide, and a cap on the number of applicants to be accepted from intra-county regions, in order to ensure that admissions are not dominated by more prosperous regions.

This example is based on the admissions system of a real high school: Thomas Jefferson High School for Science and Technology in Alexandria, Virginia, USA (see [Fai25]). The hierarchical caps

imposed on admissions can then be modeled as a laminar matroid. In order to fulfill the requirements of a lottery system, each acceptable applicant should have a fair chance of admission. If the high school were then to employ rolling admissions, where some applicants must receive a decision before others apply, we then have an online setting which is exactly the fair matroid selection problem on laminar matroids, for which the algorithm we are about to describe provides a solution.

The following algorithm assumes that when an element $e$ is presented, we learn the set of indices $j$ for which $e$ is contained in $S_j$. This is analogous to graphic matroids, where when an element, which corresponds to an edge in a graph, is presented, we learn the identities of its endpoints; it is also motivated by the previously described example of high school admissions, as we would be aware of each applicant's place of residence. The algorithm continues to be oblivious other than knowing $\alpha$ – it does not know $m$ or any of the $r_j$ upfront. The fairness achieved by this algorithm is expressed in

---

**Algorithm 2:** Algorithm for adversarial order fair matroid selection on laminar matroids.

---

Let $T$ be an ordering of the presented elements. Initially, $T$ is empty.
Let $c$ be an assignment of colors to the presented elements. Initially, $c$ is empty.
Choose $r$ uniformly at random from $1, \ldots, 2\alpha - 1$.
**for** $(e, J)$ *being presented, where $J$ contains all indices of sets containing $e$* **do**
    Define $j \in J$ so that we have previously seen an element contained in $S_j$, and among such $j$,
      the number of elements we have seen contained in $S_j$ is minimized.
    **if** $j$ *exists* **then**
      |  Insert $e$ in $T$ after the last element in $T$ contained in $S_j$.
    **else**
      |  Append $e$ to the end of $T$.
    Let $N$ be the union of the nearest $\alpha - 1$ elements to the left of $e$ in $T$ and the nearest $\alpha - 1$
      elements to the right of $e$ in $T$.
    Choose $c_e$ to be the least color among $1, \ldots, 2\alpha - 1$ that is not the color of an element of $N$.
    **if** $c_e = r$ **then**
      |  Accept $e$.

---

the below theorem and corollary, and is optimal up to a constant factor of 2.

**Theorem 11.** *Algorithm 2 is an $\alpha$-knowing algorithm that is $(2\alpha - 1)$-fair for adversarial order fair matroid selection on laminar matroids.*

**Corollary 3.** *There exists a totally oblivious algorithm for adversarial order fair matroid selection on laminar matroids which is $\tilde{O}(\alpha)$-fair. More generally, for any feasible $h$, there exists a totally oblivious algorithm for adversarial order fair matroid selection on laminar matroids which is $2h(\alpha)$-fair.*

Algorithm 2 again uses the first key idea from Algorithm 1 of converting the fair matroid selection problem to an online coloring problem; it differs by a more sophisticated method of coloring. Specifically, the algorithm maintains an ordering of the elements of the laminar matroid, with the essential property that for any set $S_j$, the elements of $S_j$ appear contiguously in the ordering. Algorithm 2 then maintains the invariant that any $\alpha$ consecutive elements in the ordering have different colors; it does this online using $2\alpha - 1$ colors, where offline it can be done trivially using $\alpha$ colors by cycling through colors. This invariant then guarantees that the set of elements of any particular color satisfies the constraint imposed by each $S_j, r_j$, ensuring the validity of the algorithm.

The full proofs of Theorem 11 and Corollary 3 are located in Appendix C.4.

## 4 Random Order Fair Matroid Selection

In this section, we consider a variant of fair matroid selection in which the elements arrive in a random order. A practical motivation of the random order setting is the following situation. We have access to a set of candidates of which we have no prior knowledge. We are then free to interview each candidate in any order that we choose; immediately after interviewing a candidate, we must choose whether or not to accept said candidate. Our goal is then to accept candidates fairly, subject to a matroid constraint on which candidates can be accepted. As we choose the order in which we interview candidates, we could always choose to interview them in a uniformly random order; on the

other hand, assuming that the candidates are chosen and arranged adversarially, the adversary could always choose to arrange the candidates uniformly randomly. Thus, this model is equivalent to the candidates being fixed to arrive in a random order, with the added detail that the algorithm is allowed to know the number $n$ of arriving elements beforehand.

We therefore first formally define the random order fair matroid selection problem as follows: a matroid $M = (E, \mathcal{I})$ is chosen, unknown to the algorithm. The algorithm is first told $n = |E|$, then sees the elements of $E$ in a uniformly random order, which the algorithm must accept or deny online, subject to an independence constraint. Like before, the algorithm can access the matroid online via independence queries containing only elements which it has already seen. The algorithm is then judged based on its fairness, as usual. We will describe algorithms that rely on no upfront knowledge other than $n$ as simply *oblivious*, while algorithms with upfront knowledge will be described as, for example, $\alpha$-knowing, as before.

We now note that for the purpose of algorithm design, rather than working directly in the above model, it may be more natural to work in a *sampling model*. In the sampling model, the algorithm specifies probabilities $p_1, \ldots, p_m$ that sum to less than 1; then, the elements of $E$ are partitioned into sets $S_1, \ldots, S_m, T$, each being placed independently into $S_j$ with probability $p_j$ (with $T$ collecting the remainder of elements). The algorithm is first presented with the sets $S_1, \ldots, S_m$ without the possibility of accepting their elements, then sees $T$ in a random order and must accept or deny its elements online.

It can be shown that an algorithm working in the sampling model implies an algorithm for the standard model of random order fair matroid selection; a formal proof of this is located in Appendix C.5 in addition to a formal definition of the sampling model. As an example of the application of the sampling model, we note that it allows us to approximate the arboricity $\alpha$ up to a constant factor – roughly, we can sample half of the elements, then multiply their arboricity by 2 to obtain a proxy for the true arboricity of the matroid (the full proof is deferred to Appendix C.6).

**Theorem 12.** *Suppose that for some class $C$ of matroids closed under restriction, we are given an algorithm $A_\alpha$ parameterized by the arboricity $\alpha$ and a nondecreasing $f : \mathbb{Z}_+ \to \mathbb{R}$ that grows polynomially, such that the algorithm $A_\alpha$ is $f(\alpha)$-fair when the input matroid $M \in C$ is $\alpha$-arboric. Then there exists an oblivious algorithm $A'$ for random order fair matroid selection that is $O(f(\alpha))$-fair on $M \in C$.*

Thus, unlike the adversarial order model, there is no logarithmic loss necessary to remove foreknowledge of $\alpha$. The rank $k$ can be approximated similarly (see Appendix C.7). Together, these imply that the $\alpha$-knowing and $(\alpha, k)$-knowing algorithms derived in Section 3 automatically extend to random order fair matroid selection. We now move to the algorithm unique to random order fair matroid selection, which achieves $O(\alpha)$-fairness on the class of graphic matroids.

## 4.1 Graphic matroids

We now focus on *graphic matroids*, which are the class of matroids where the independent sets are forests in some graph $G$. An equivalent definition for a set $S$ of edges being a forest (i.e. independent in the graphic matroid) is that there is some ordering of the vertices of $G$ such that for each vertex $v \in V$, there is at most one edge in $S$ connecting $v$ to a vertex earlier in the ordering. A natural strategy for ensuring that we select an independent subset of edges is then to pick such an ordering, group edges by the endpoint of theirs that appears later in the ordering, and pick at most one such edge. Indeed, this is the basis for a simple $2e$-competitive algorithm for graphic matroid secretary [KP09] – we select this ordering randomly, then for each group, simply play the normal secretary problem, attempting to select the highest weight edge.

The trouble for the fair matroid selection problem is that we are not simply attempting to select the highest weight edge – rather, we would like to select edges fairly, meaning that all edges have some probability of being selected. One way we might hope to do this is to select an ordering of the vertices such that each vertex $v$ has at most $\alpha$ edges connecting it to vertices earlier in the ordering; this brings us to the concept of *degeneracy*.

**Definition 3.** *Given a nonnegative integer $d$, an undirected graph $G = (V, E)$ is $d$-degenerate if there exists an ordering $v_1, \ldots, v_{|V|}$ of $V$ such that for any $1 \leq j \leq |V|$, there exist at most $d$ edges in $E$ of the form $(v_i, v_j)$ for $i < j$. The degeneracy $d$ of $G$ is the minimum $d$ such that $G$ is $d$-degenerate.*

In general, the degeneracy $d$ is not guaranteed to be equal to the arboricity $\alpha$. However, crucially for us, they are within a constant factor of each other as shown by the following well-known theorem:

**Theorem 13.** *Given an undirected graph $G$ with arboricity $\alpha$ and degeneracy $d$, we have $\alpha \leq d < 2\alpha$.*

We are thus guaranteed that for $d < 2\alpha$, there exists an ordering as we desire parameterized by $d$. However, the question still remains of how to compute such an ordering. Naturally, we cannot compute such an ordering without knowing anything about the graph, meaning that we would like to learn this ordering based on a sample of the edges. In order to facilitate this, we will introduce the concept of a *d-level* of vertices in a graph. For reasons that will become clear later, we will define the $d$-level for a *directed* graph.

**Definition 4.** *Let $G = (V, E)$ be a directed graph. Then we will define the d-level, $l_d : V \to \mathbb{N}_0 \cup \{\infty\}$, via the following iterative process. Number iterations starting from 0; in the j-th iteration, we will remove all vertices of $G$ with outdegree at most $d$, and set their d-levels to be $j$. The process ends when no remaining vertices with outdegree at most $d$; those vertices will have a d-level of $\infty$.*

Our algorithm will work by computing $l_d$ on a sample of the edges, which will give rise to an ordering, from which we can apply the previously described idea of grouping edges by their later endpoint. The following theorem describes various properties of the $d$-level in order to facilitate better understanding of it; a formal proof of these properties is deferred to Appendix C.8.

**Theorem 14.** *Let $G = (V, E)$ be an undirected graph, and define its d-level $l_d$ by repeating each edge of $G$ in each direction to obtain a directed graph. Then $G$ is d-degenerate iff $l_d$ is never $\infty$. Furthermore, if $G$ is d-degenerate, then we can obtain an appropriate ordering of $V$ by sorting the vertices in decreasing order of d-level.*
*Additionally, if $G$ is d-degenerate, then $l_d$ is the unique function with the following properties: first, a vertex $v \in V$ with degree at most $d$ has $l_d(v) = 0$; second, for a vertex $v \in V$ with degree larger than $d$, let $u$ be the neighbor of $v$ with the $d + 1$-th highest d-level; then $l_d(v) = l_d(u) + 1$.*

Having established the concept of the $d$-level in a graph, we now move to the actual algorithm, which will work in the framework of Theorem 18, taking multiple independent samples of edges. The

---

**Algorithm 3:** Algorithm for random order fair matroid selection on $d$-degenerate graphs.

---

Define $q = \frac{1}{3}d^{\frac{1}{4}}$.
Let $m = 2$ and $p_1, p_2 = \frac{1-q}{2}$.
Let $S_1, S_2$ be the presented samples of edges.
Initialize a directed graph $H$ with no edges.
**for** $(u, v) \in S_1$ *with $u < v$* **do**
  | Insert $u \to v$ into $H$.
**for** $(u, v) \in S_2$ *with $u < v$* **do**
  | Insert $v \to u$ into $H$.
Let $U$ be an ordering of the vertices in $H$ by decreasing $l_d$, breaking ties by vertex label.
**for** $e = (u, v)$ *being presented after the samples* **do**
  | **for** $w \in \{u, v\}$ *such that $w \notin U$* **do**
  |  | Set $l_d(w)$ to 0.
  |  | Insert $w$ into $U$ appropriately.
  | Order $(u, v)$ as $(a, b)$ where $a$ appears before $b$ in $U$.
  | If no edge with later endpoint $b$ has been accepted yet, accept $e$.

---

fairness achieved by this algorithm is expressed in the below theorem and corollary.

**Theorem 15.** *When given an integer $d \geq 3$, Algorithm 3 is $(2 + o(1))d$-fair in the random order fair matroid selection setting on graphic matroids where the graph is $d$-degenerate.*

**Corollary 4.** *There exists an $\alpha$-knowing algorithm for random order fair matroid selection on graphic matroids which is $(4 + o(1))\alpha$-fair. Furthermore, there exists an oblivious algorithm for random order fair matroid selection on graphic matroids which is $O(\alpha)$-fair.*

Most of the intuition behind Algorithm 3 has already been described in the previous discussion of ordering, degeneracy, and $d$-level. The key remaining component is that we do not simply sample each edge to produce an $H$ whose $d$-level we compute; rather, we convert $G$ to a directed graph by

sampling each undirected edge $(u, v)$ as either the directed edge $u \to v$ or the directed edge $v \to u$ – this double sampling gives an intuition as to the reason for the constant 2 in the fairness. More correctly, we simulate this process by directing edges in $S_1$ as $u \to v$ and edges in $S_2$ as $v \to u$. The motivation for this is that for a vertex $v$, we can fix all edges which are not outgoing from $v$, which do not directly affect the $d$-level of $v$, then reason about which outgoing edges of $v$ are sampled into $H$, which directly affects the $d$-level of $v$ but not any other vertices.

Naturally, these still indirectly affect all $d$-levels, as they may rely on the $d$-level of $v$. We handle this in the analysis, by introducing an additional notion referred to as the "$v$-fixed $d$-level"; this is defined in the same way as the usual $d$-level, except that $v$ is fixed to never be removed by the algorithm, and thus has a $v$-fixed $d$-level of $\infty$. We then show firstly, that the $v$-fixed $d$-level does not depend on which edges are outgoing from $v$; and secondly, that all $v$-fixed $d$-levels of neighbors of $v$ except for the $d$ largest are equal to the corresponding simple $d$-levels. These properties allow analyzing the $d$-level independently from the edges outgoing from $v$ by using the $v$-fixed $d$-level as a proxy.

The proofs of Theorem 15 and Corollary 4 are located in Appendix C.9.

## Acknowledgements

The work is partially supported by DARPA QuICC, ONR MURI 2024 award on Algorithms, Learning, and Game Theory, Army-Research Laboratory (ARL) grant W911NF2410052, NSF AF:Small grants 2218678, 2114269, 2347322.

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

## A    Further Related Work

**Online selection problems**    Our work is closely related to research on prophet inequalities and the secretary problem. In their classical formulations, a sequence of values arrives online, and the goal is to irrevocably select a large element. In the secretary problem, the set of values is chosen adversarially, but their arrival order is random. Conversely, in prophet inequalities, the arrival order is adversarial, but each element is drawn from a known distribution. These problems have been extensively studied over the past two decades, with many generalizations to settings such as matroids, matchings, and combinatorial auctions [KW12, Ala14, FGL14, Rub16, DFKL20, EFGT22a].

When the distributions are known and items arrive in random order, the problem is referred to as the prophet secretary problem, first introduced by Esfandiari et al. [EHLM17], who designed an algorithm with a competitive ratio of $1 - 1/e$. Subsequent works have improved this ratio [ACK18, CSZ19, CSZ21], extended the problem to matroids and combinatorial auctions [EHKS18], and obtained stronger results for the special case of i.i.d. variables [AEE$^+$17, CFH$^+$17, PT22].

While the original version of the secretary problem assumes that the elements have weights, one can consider a variant where the elements are merely ordered. This uniquely determines the maximal independent set, allowing for various notions of "competing with the optimum" [STV21]. One such notion is *probability competitiveness*; an algorithm is said to have a probability competitive ratio of $c \geq 1$ if each element of the offline optimum appears in the output with probability at least $c^{-1}$. For general matroids, the best known probability competitive ratio is $O(\log k)$. Interestingly,

in our problem, we achieve a similar $O(\log k)$ approximation to the offline optimum, despite the significantly more challenging adversarial arrival model (see Theorem 10).

Traditionally, algorithms for prophet inequalities and secretary problems have relied on pricing-based approaches. More recently, however, new techniques based on Online Contention Resolution Schemes (OCRS) have emerged [EFGT22b, EFGT22a, MMG24]. Originally introduced by Feldman et al. [FSZ16], the classical OCRS problem is formulated as follows: a sequence of elements arrives in random order, and each element reveals whether it is active upon arrival. Activeness is determined independently based on draws from a pre-specified Bernoulli distribution, where the parameters are chosen to ensure that, on average, the active elements satisfy a given combinatorial constraint—e.g., for matroids, the parameter vector lies in the matroid polytope. Given a parameter $c \in (0, 1)$, the goal in the $c$-OCRS problem is to accept a subset of active elements such that, conditioned on being active, each element is selected with probability at least $c$. Later works established the equivalence of OCRS with prophet inequalities [LS18] and extended it to the random arrival model [AW18].

While the original formulation in Feldman et al. [FSZ16] assumes independence in the activeness of elements, allowing correlations across elements makes the problem significantly more challenging. For matroids, obtaining constant approximations to the offline optimum is equivalent to matroid secretary as shown by Dughmi [Dug19, Dug21]. Crucially, as with standard OCRS, the activeness distribution must be known in advance.

While our work shares similarities with OCRS in seeking a lower bound on the minimum probability of acceptance, it differs in two key aspects. First, we make no distributional assumptions on the input, allowing for a fully adversarial setting. Second, we derive results for both random and adversarial arrival orders. Notably, for correlated distributions, Dughmi [Dug21] showed that no meaningful guarantees can be achieved if the order is adversarial, or the input distribution is unknown.

**Fairness**  Fair division is commonly studied in game theory, with the roots going as far back as 1948 [Ste48]. Different notions of fairness are considered in the literature including max-min fairness, Nash welfare, and envy-freeness. A notable example of max-min fairness in allocation problems is the Santa Claus problem which seeks to distribute $m$ items among $n$ agents in a way that maximizes the minimum utility. The general version of the problem was originally studied under a different name by Lipton et al. [LMMS04]. Later, Bezáková and Dani [BD05] obtained an algorithm achieving $\frac{\log \log m}{\log \log \log m}$ approximation for the special case of restricted assignment. Many works have since studied the problem from a computational perspective [Fei08, HSS11, AFS12, PS15, AKS17, CM19]. Notably, Haeupler et al. [HSS11] provide a constant approximation for the problem. Recently, Springer et al. [SHPK22] studied the online version of the Santa Claus problem. While the notion of max-min fairness is also used in our formulation, our setting is fundamentally different. Existing work on the Santa Claus problem focus on distributing items among agents; in contrast, we assume that the agents themselves are accepted under a matroid constraint and focus on minimizing the minimum (expected) agent utility.

Recent work by Balkanski, Ma, and Maggiori [BMM24] considered a notion of fair secretaries. They build upon work, also recent, by Fujii and Yoshida [FY23] showing how in the classical secretary setting, an algorithm that is given predictions about the arriving elements can attain a competitive ratio approaching 1. Balkanski, Ma, and Maggiori contend that these predictions, if biased, could cause the probability of selecting the maximum element to become 0, which they view as unfair, and propose algorithms to maintain fairness toward the maximum element while still allowing the competitive ratio to tend to 1. This is mostly unrelated to our work, as we consider a different notion of fairness that aims to provide a decent probability to all elements rather than ensure that a maximum element is properly rewarded; furthermore, our setting does not have weights or predictions, and allows multiple elements to be selected according to a matroid constraint.

# B  Fair Matchoid Selection

In this section, we generalize the fair matroid selection problem by considering imposing multiple matroid independence constraints on the accepted set. A well-studied structure for combining multiple matroid constraints is the $p$-matchoid, which is defined as follows: we have a universe $U$ of elements, and $m$ matroids $M_1, \ldots, M_m$ such that for each $M_j = (E_j, \mathcal{I}_j)$, the element set $E_j$ is a subset of $U$. Furthermore, we limit each element $u \in U$ to be contained in at most $p$ of these matroids, i.e. there

are at most $p$ different $j$ such that $e \in M_j$. A subset $A \subseteq U$ satisfies the $p$-matchoid constraint if for all $j$, $A \cap E_j \in \mathcal{I}_j$.

We first note that, unlike the single matroid case, where the integer parameter of arboricity allowed us to cleanly characterize the optimal fairness up to an additive factor of $1$, due to Edmonds' generalization of the Nash-Williams theorem (Theorem 6). However, this is not the case for $p$-matchoids. Specifically, given a $p$-matchoid, define the matchoid rank $r : 2^U \to \mathbb{N}_0$ so that $r(A)$ is the size of the largest subset of $A$ that satisfies the matchoid constraint. Then let $c_{\text{density}} = \max_{A \subseteq U} \frac{|A|}{r(A)}$, let $c_{\text{partition}}$ be the minimum number of sets satisfying the matchoid constraint into which we can partition $U$, and let $\beta$ be the infimum value such that there exists an offline $\beta$-fair algorithm for fair matroid selection on the $p$-matchoid. It can then be shown that $c_{\text{density}} \leq \beta \leq c_{\text{partition}}$. However, it can also be shown that it is possible for $\frac{\beta}{c_{\text{density}}} = \Omega(\lg \lg p)$ to hold, and that it is possible for $\frac{c_{\text{partition}}}{\beta} = \Omega(\lg \lg p)$ to hold.

This definition of matchoid rank is a natural generalization of the rank in matroids, and viewed as such, Theorem 6 states that $\lceil c_{\text{density}} \rceil = c_{\text{partition}}$. As this equality fails in the more general matchoid case, we can no longer characterize the optimal fairness by an "arboricity." Thus, in the absence of a natural characterization, we will rely directly on the optimal fairness to describe our algorithm. As in the preceding paragraph, we will use $\beta$ to refer to the optimal fairness achievable offline for a given $p$-matchoid. In spite of this, the quantity $c_{\text{density}}$ will be relevant in the analysis of the algorithm we describe below.

We first note that Theorem 8 can be adapted to the $p$-matchoid case without issue; indeed, said theorem applies more generally to any problem where randomizing over a set of algorithms yields performance at least the expected performance of each individual algorithm.

**Theorem 16.** *Suppose that for some class $C$ of matchoids, we have a function $f : C \to \mathbb{Z}_+$ defining a property of matchoids in $C$. Further suppose that we are given an algorithm $A_k$ parameterized by a positive integer $k$ and a constant $c \in \mathbb{R}_+$ such that the algorithm $A_k$ is $ck$-fair when the input matchoid $M \in C$ satisfies $f(M) \leq k$. Then for any feasible $h$, there exists a totally oblivious fair matchoid selection algorithm $A'$ which is $c \cdot h(f(M))$-fair on $M \in C$.*

*Proof.* The proof is identical to the proof of Theorem 8. $\qquad\square$

We now introduce and analyze the below algorithm, which extends our algorithm for general matroids to the $p$-matchoid case. The performance of the above algorithm is described in the below theorem

---

**Algorithm 4:** General algorithm for adversarial order fair matchoid selection.

---

Let $m = \lceil \beta(p \ln k + 1) \rceil$.
For each $j = 1, \ldots, m$, initialize the set $A_j$ as empty.
Choose $r$ uniformly at random from $1, \ldots, m$.
**for** *e being presented* **do**
    Let $j$ be the minimum among $1, \ldots, m$ such that $A_j \cup \{e\}$ satisfies the matchoid constraint.
    Insert $e$ into $A_j$.
    **if** $j = r$ **then**
        Accept $e$.

---

and corollary:

**Theorem 17.** *Given a $p$-matchoid, let $\beta$ be the optimal fairness achievable for said $p$-matchoid and let $k$ be the size of a maximum set satisfying the matchoid constraint. Then Algorithm 4 is an $\lceil \beta(p \ln k + 1) \rceil$-knowing, $\lceil \beta(p \ln k + 1) \rceil$-fair algorithm for adversarial order fair matchoid selection.*

**Corollary 5.** *There exists a totally oblivious algorithm (i.e. oblivious even to $p$) algorithm for adversarial order fair matchoid selection which is $\tilde{O}(p\beta(\ln k + 1))$-fair.*

*Proof.* Let $C$ be the class of all finite matchoids, and define $f : C \to \mathbb{Z}^+$ by $f(M) = \lceil \beta(p \ln k + 1) \rceil$. Then we can apply Theorem 16 to Algorithm 4 and the loss function $h(j) = 2j \log^2 j$ (as in Corollary 1) to obtain the desired result. $\qquad\square$

Algorithm 4 is in fact almost identical to Algorithm 1; the key additional complication is that while the first set $A_1$ is still guaranteed to be maximal, a maximal set satisfying a matchoid constraint is not necessarily a maximum set satisfying the matchoid constraint. We therefore must additionally use the key property that a maximal set in a $p$-matchoid has size at least $\frac{1}{p}$ times the size of a maximum set in the $p$-matchoid. An additional complication comes from the fact that we must compare the fairness of Algorithm 4 to the optimal fairness $\beta$, rather than using a natural parameter such as $\alpha$. We handle this by using the fact that $c_{\text{density}} \leq \beta$ still holds as an inequality.

We now analyze Algorithm 4.

*Proof of Theorem 17.* The structure of the proof is similar to that of the proof of Theorem 10; we first note as in that proof, that it suffices to show that all elements of the $p$-matchoid $M$ are contained in one of $A_1, \ldots, A_m$; given this, the algorithm will successfully execute ($j$ will always be well-defined), and each element $e$ will be taken with probability $\frac{1}{m}$ as it will fall into some $A_j$, and we will choose $r = j$ with probability $\frac{1}{m}$.

We now define sets $S_0, \ldots, S_m$ (analogously to the single matroid case) where $S_j$ contains all elements of the $p$-matchoid $M$ not contained in any of $A_1, \ldots, A_j$. Our goal is then to show that $S_m$ is empty. Given this, we state the following lemma; we omit its proof as it is essentially the same as the proof of Lemma 4.

**Lemma 1.** *For each $j = 1, \ldots, m$, $A_j$ is a maximal set satisfying the matchoid constraint within the restriction of the $p$-matchoid to $S_{j-1}$.*

The key differences come in the below lemma, where we must utilize both the property of a $p$-matchoid that a maximal set $p$-approximates a maximum set, and the inequality $c_{\text{density}} \leq \beta$.

**Lemma 2.** *For each $j = 1, \ldots, m$, $|A_j| \geq \frac{1}{p\beta}|S_{j-1}|$.*

*Proof.* We will first show that there exists a subset $A'$ of $S_{j-1}$ satisfying the matchoid constraint such that $|A'| \geq \frac{1}{\beta}|S_{j-1}|$. To see this, note that $\beta \geq c_{\text{density}} = \max_{X \subseteq U} \frac{|X|}{r(X)}$, where $r(X)$ is the size of the maximum subset of $X$ satisfying the matchoid constraint. This then implies that $c_{\text{density}} \geq \frac{|S_{j-1}|}{r(S_{j-1})}$, meaning that $r(S_{j-1}) \geq \frac{|S_{j-1}|}{c_{\text{density}}} \geq \frac{|S_{j-1}|}{\beta}$, and so there exists some $A' \subseteq S_{j-1}$ satisfying the matchoid constraint with $|A'| \geq \frac{1}{\beta}|S_{j-1}|$ as desired.

We invoke Lemma 1 to see that $A_j$ is a maximal subset of $S_{j-1}$ satisfying the matchoid constraint, meaning that by the well-known property of $p$-matchoids that any maximal set has size at least $\frac{1}{p}$ that of a maximum set, it must be that $|A_j| \geq \frac{1}{p}|A'| \geq \frac{1}{p} \cdot \frac{1}{\beta}|S_{j-1}|$ as desired. $\qquad\square$

A final difference from the proof of Theorem 10 is that as we are no longer relying on the arboricity, it is not immediately obvious that the number, $n$, of elements in the matchoid is at most $\beta k$. However, this can still be shown by again appealing to the inequality $c_{\text{density}} \leq \beta$: as $c_{\text{density}} = \max_{X \subseteq U} \frac{|X|}{r(X)}$, we specifically note that for the whole universe, $\beta \geq c_{\text{density}} \geq \frac{|U|}{r(U)} = \frac{|U|}{k}$, immediately implying that $n = |U| \leq \beta k$.

As the remainder of the proof follows the same path as the proof of Theorem 10, we will more concisely describe it. The set $S_0$ has $n \leq \beta k$ elements. By Lemma 2, each set $A_j$ removes an at least $\frac{1}{p\beta}$ fraction of the elements that were in $S_{j-1}$. It follows that after $m' = \lceil -\log_{1-\frac{1}{p\beta}} k \rceil$ steps, we have at most $\frac{n}{k} \leq \beta$ elements remaining, i.e. $|S_{m'}| \leq \beta$, and so because cardinalities are integers, we have that $|S_{m'}| \leq \lfloor \beta \rfloor$. At that point, by Lemma 1 it is naturally still true that each $A_j$ removes $\geq 1$ element as long as some elements remain. Thus after at most $\lfloor \beta \rfloor$ additional steps, all elements are exhausted, and so $S_{m'+\lfloor \beta \rfloor}$ must be empty.

We now complete the proof by noting as in the proof of Theorem 10 that $-\log_{1-\frac{1}{p\beta}} k \leq p\beta \ln k$, and so $m' + \lfloor \beta \rfloor \leq \lceil p\beta \ln k \rceil + \lfloor \beta \rfloor \leq m$, meaning that $S_m$ must be empty as desired.

$\qquad\square$

## C  Omitted proofs

### C.1  The optimal offline fairness is essentially the arboricity

*Proof of Theorem 7.* We first prove that there exists an $\alpha$-fair offline algorithm for fair matroid selection on $M$. By the definition of $\alpha$, there exists a partition of $E$ into $\alpha$ sets $A_1, \ldots, A_\alpha$ such that each $A_j$ is independent. We can therefore sample $j$ uniformly at random from $1, \ldots, \alpha$, then select all elements in $A_j$. As each element is in exactly one such set, each element is selected with probability $\frac{1}{\alpha}$. Furthermore, as each such set is independent, the algorithm is valid. Thus, this algorithm is $\alpha$-fair on $M$.

We now prove that no $(\alpha - 1)$-fair offline algorithm exists for fair matroid selection on $M$. Suppose that such an algorithm $A$ did exist; then it selects each element of $M$ with probability at least $\frac{1}{\alpha-1}$. By Theorem 6, there exists a set $S \subseteq E$ such that $\alpha = \lceil \frac{|S|}{\operatorname{rank} S} \rceil$; it follows that $\alpha - 1 < \frac{|S|}{\operatorname{rank} S}$. Now define a random variable $X$ equal to the expected number of elements of $S$ selected by $A$. On one hand, each element is selected with probability at least $\frac{1}{\alpha-1}$, so $X \geq \frac{|S|}{\alpha-1}$. On the other hand, the set of selected elements must be independent, meaning that at most $\operatorname{rank} S$ elements can be selected from $S$, and so $X \leq \operatorname{rank} S$. We thus have that $\operatorname{rank} S \geq \frac{|S|}{\alpha-1}$. This contradicts our earlier assertion that $\alpha - 1 < \frac{|S|}{\operatorname{rank} S}$, and so we are done.

### C.2  Adapting algorithms to be oblivious in the adversarial order setting

*Proof of Theorem 8.* We will define $A'$ by randomly choosing $l$, then running $A_l$. If at some point $A_l$ attempts to make an invalid choice (i.e. select an element which would cause its set of selected elements to be dependent), we instead terminate, not accepting any further elements. Let $p_l$ be the probability that $A'$ runs $A_l$. We will set $p_l = l[\frac{1}{h(l)} - \frac{1}{h(l+1)}]$. We must first show that $\sum_{l=1}^{\infty} p(l) \leq 1$, so that the random selection is well defined. We can see this via a telescoping sum:

$$\sum_{l=1}^{\infty} p(l) = \sum_{l=1}^{\infty} l \left[ \frac{1}{h(l)} - \frac{1}{h(l+1)} \right] = \sum_{l=1}^{\infty} \frac{1}{h(l)} [l - (l-1)] = \sum_{l=1}^{\infty} \frac{1}{h(l)} \leq 1.$$

Now, let $M$ be some matroid in $C$. We want to show that $A'$ is $c \cdot h(f(M))$-competitive on $M$. To see this, first note that for any $l \geq f(M)$, $A_l$ is $cl$-fair on $M$. Therefore, any element $e \in M$ is selected by $A_l$ with probability at least $\frac{1}{cl}$. It follows that $e$ is selected by $A'$ with probability at least the expected value of $b_l$ under the random selection of $l$, where $b_l = \frac{1}{cl}$ for $l \geq f(M)$ and $b_l = 0$ for $l < f(M)$. This can again be computed by a telescoping sum:

$$E[b_l] = \sum_{l=1}^{\infty} p_l b_l = \sum_{l=f(M)}^{\infty} l \left[ \frac{1}{h(l)} - \frac{1}{h(l+1)} \right] \cdot \frac{1}{cl} = \frac{1}{c} \sum_{l=f(M)}^{\infty} \left[ \frac{1}{h(l)} - \frac{1}{h(l+1)} \right] = \frac{1}{c \cdot h(f(M))}.$$

This completes the proof. $\qquad\square$

*Proof of Theorem 9.* In a rank-1 matroid, all elements are indistinguishable, and $A$ can only select an element if it has not selected any prior, so the only information $A$ has when deciding whether or not to select an element is the amount of elements seen so far. Thus, $A$ is defined by the probability $p_j$ with which it selects the $j$-th element it is shown. Because $A$ can select at most 1 element, it must be that $\sum_{k=1}^{\infty} p_j \leq 1$.

Then, because the $j$-th element is selected with probability $p_j$, the minimum probability with which any of the first $j$ elements is selected is at most $p_j$; since the set of the first $j$ elements has arboricity $j$, $A$ is then at best $\frac{1}{p_\alpha}$-fair on matroids of arboricity $j$. Thus, $h(\alpha) \geq \frac{1}{p_\alpha}$, and so $\sum_{l=1}^{\infty} \frac{1}{h(l)} \leq 1$. Furthermore, $(\alpha - 1)$-arboric matroids are also $\alpha$-arboric, so it must be that $h(\alpha - 1) \leq h(\alpha)$; thus, $h$ is feasible. $\qquad\square$

*Proof of Corollary 1.* We first show that the function $h(j) = 2(j+2)\log^2(j+2)$ is feasible.

Observe that, since $h(j)$ is increasing, the sum $\sum_{j=1}^{\infty} \frac{1}{h(j)}$ is at most

$$\sum_{j=1}^{\infty} \frac{1}{h(j)} = \sum_{j=3}^{\infty} \frac{1}{2j \log^2(j)} \leq \frac{1}{2} \int_{2}^{\infty} \frac{dx}{x \log^2(x)}.$$

Given that, calculate the integral. Using the substitution $u = \log x$, so that $du = \frac{dx}{x}$, we rewrite it as

$$\int_{2}^{\infty} \frac{dx}{x \log^2 x} = \int_{\log 2}^{\infty} \frac{du}{u^2}.$$

Since $\int u^{-2}\, du = -\frac{1}{u} + C$, evaluating from $u = \log 2$ to $u \to \infty$ gives $\left[-\frac{1}{u}\right]_{\log 2}^{\infty} = \frac{1}{\log 2}$. Therefore, $\sum_{j=1}^{\infty} 1/h(j) \leq 1$ as desired.

Given that $h$ is feasible, we can then apply Theorem 8 using $h$. $\qquad\square$

### C.3 Analysis of Algorithm 1

*Proof of Theorem 10.* We first note that the theorem holds as long as Algorithm 1 is able to correctly execute.

**Lemma 3.** *If Algorithm 1 is able to correctly execute, that is, for each element $e$, the index $j$ is well-defined, then Algorithm 1 is a valid $\lceil \alpha(\ln k + 1)\rceil$-fair algorithm for adversarial order fair matroid selection.*

*Proof.* Supposing that Algorithm 1 is able to correctly execute, then the set of elements it selects will be precisely one of the final sets $A_1, \ldots, A_m$ chosen at random. Each set is guaranteed by the way the algorithm chooses $j$ to be independent, meaning that the selection of the algorithm is valid. Furthermore, every element is in exactly one set, meaning that each element is selected with probability exactly $\frac{1}{m}$. Therefore, $Algorithm\ 1$ is $m$-fair, and so $\lceil \alpha(\ln k + 1)\rceil$-fair. $\qquad\square$

We now proceed to show that Algorithm 1 correctly executes. It suffices to show that all elements of $M$ are contained in one of $A_1, \ldots, A_m$. We will prove this via the below lemmas. We first, for each $j = 0, \ldots, m$, define $S_j$ to be all elements of $M$ that are not contained in any of $A_1, \ldots, A_j$. In particular, $S_0$ consists of all elements of $M$, and we would like to prove that $S_m$ is empty.

**Lemma 4.** *For each $j = 1, \ldots, m$, $A_j$ is a maximal independent set within the submatroid $S_{j-1}$.*

*Proof.* We have already seen that $A_j$ is an independent set. We can see that it is maximal by contradiction. Suppose that there is some element $e$ in $S_{j-1}$ not contained in $A_j$ such that $A_j \cup \{e\}$ is independent. As $e \in S_{j-1}$, $e$ is also not contained in any of $A_1, \ldots, A_{j-1}$. Therefore, at the time that $e$ was presented, the minimum $l$ such that $A_l \cup \{e\}$ was independent was greater than $j$ (otherwise, $e$ would have been placed in one of $A_1, \ldots, A_j$). This implies that $A_l \cup \{e\}$ was not independent for any $l = 1, \ldots, j$, and so $A_l \cup \{e\}$ is still not independent for any $l = 1, \ldots, j$, as the sets $A_l$ only have elements added to them. This contradicts the fact that $A_j \cup \{e\}$ is independent, so we are done. $\qquad\square$

We now make use of the above lemma to show how the sizes of the sets $S_j$ must be geometrically decreasing.

**Lemma 5.** *For any $j = 0, \ldots, m$, $|S_j| \leq (1 - \frac{1}{\alpha})^j n$, where $n = |S_0|$ is the number of elements in $M$.*

*Proof.* Proof by induction. The base case $j = 0$ simply states that $|S_0|$ has at most $n$ elements; as $S_0$ has exactly $n$ elements this is immediate. We would then like to show for $j \geq 1$ that assuming $|S_{j-1}| \leq (1 - \frac{1}{\alpha})^{j-1} n$, it follows that $|S_j| \leq (1 - \frac{1}{\alpha})^j n$. It suffices to show that $|S_j| \leq (1 - \frac{1}{\alpha})|S_{j-1}|$. To see this, note that $S_{j-1} = A_j \sqcup S_j$, and so $|S_{j-1}| = |A_j| + |S_j|$. Now, let $k'$ be the rank of $S_{j-1}$. By Lemma 4, $A_j$ is a maximal independent set within $S_{j-1}$; this implies that $A_j$ has rank, and therefore cardinality, equal to $k'$. However, recall that $M$ has arboricity $\alpha$, meaning that it can be partitioned into $\alpha$ independent sets. We can see by taking the intersection of these sets with

$S_{j-1}$ that $S_{j-1}$ itself can be partitioned into $\alpha$ independent sets. Each such set would then have cardinality at most $k'$, meaning that $S_{j-1}$ must have cardinality at most $\alpha k'$. We therefore have that $|A_j| = k' = \frac{1}{\alpha} \cdot \alpha k' \geq \frac{1}{\alpha}|S_{j-1}|$. Thus, $|S_j| = |S_{j-1}| - |A_j| \leq |S_{j-1}| - \frac{1}{\alpha}|S_{j-1}| = (1 - \frac{1}{\alpha})|S_{j-1}|$ as desired. $\qquad\square$

We then apply the above lemma to see that, defining $m' = \lceil -\log_{1-\frac{1}{\alpha}} k \rceil$, it must be that $|S_{m'}| \leq \frac{1}{k} \cdot n$. Once again recall that by the arboricity, we have that $n \leq \alpha k$, meaning that $|S_{m'}| \leq \alpha$. We finally prove the following lemma, which is essentially an easy version of Lemma 5.

**Lemma 6.** *For any $j = 0, \ldots, \alpha$, $|S_{m'+j}| \leq \alpha - j$.*

*Proof.* Proof by induction. In the base case, we have $|S_{m'}| \leq \alpha$, which we have already proven. We then must show that for $j = 1, \ldots, \alpha$, if $|S_{m'+(j-1)}| \leq \alpha - j + 1$, then $|S_{m'+j}| \leq \alpha - j$. We can see this as follows. $S_{m'+j} \subseteq S_{m'+(j-1)}$, meaning that if $|S_{m'+(j-1)}| < \alpha - j + 1$, then we are done. Otherwise, $|S_{m'+(j-1)}| = \alpha - j + 1 > 0$, as $j \leq \alpha$. This implies that $S_{m'+(j-1)}$ has nonzero rank, and so $A_{m'+j}$, which is a maximal independent set in $S_{m'+(j-1)}$ by Lemma 4, must be nonempty. It thus follows from $|S_{m'+(j-1)}| = |A_{m'+j}| + |S_{m'+j}|$ that $|S_{m'+j}| \leq |S_{m'+(j-1)}| - |A_{m'+j}| = \alpha - j + 1 - |A_{m'+j}| \leq \alpha - j + 1 - 1 = \alpha - j$ as desired. $\quad\square$

The above lemma can then be applied to see that $|S_{m'+\alpha}| \leq \alpha - \alpha = 0$. Thus, $S_{m'+\alpha}$ is empty, and so because $S_j$ contains $S_{j1}$ for all $j$, assuming that $m' + \alpha \leq m$, we will have proven that $S_m$ is empty as desired. We finally show said inequality: substituting in the values of $m'$ and $m$, it becomes $\lceil -\log_{1-\frac{1}{\alpha}} k \rceil + \alpha \leq \lceil \alpha(\ln k + 1) \rceil$. The right hand side can be written as $\lceil \alpha \ln k \rceil + \alpha$, meaning that the inequality reduces to $\lceil -\log_{1-\frac{1}{\alpha}} k \rceil \leq \lceil \alpha \ln k \rceil$, for which it suffices to show that $-\log_{1-\frac{1}{\alpha}} k \leq \alpha \ln k$. This can be expanded by properties of logarithm as $-\frac{\ln k}{\ln(1-\frac{1}{\alpha})} \leq \alpha \ln k$, which is then equivalent to $-\frac{1}{\ln(1-\frac{1}{\alpha})} \leq \alpha$. This can be reorganized as $-\frac{1}{\alpha} \geq \ln(1 - \frac{1}{\alpha})$, which then follows as an instance of the inequality $x \geq \ln(1 + x)$ for $x < 0$; this is true as it is tight at $x = 0$ and the slope of $\ln(1 + x)$ is greater than 1 for $x < 0$. $\qquad\square$

$\qquad\square$

*Proof of Corollary 2.* Let $C$ be the class of all finite matroids, and define $f : C \to \mathbb{Z}+$ by $f(M) = \lceil \alpha(M)(\ln k(M) + 1) \rceil$. Then we can combine Algorithm 1, which is $f(M)$-fair when it knows $f(M)$ beforehand, with Corollary 1 (where $c = 1$) to see that there must exist a totally oblivious algorithm for fair matroid selection that is $\tilde{O}(\lceil \alpha(\ln k + 1) \rceil) = \tilde{O}(\alpha \ln k)$-fair. $\qquad\square$

### C.4 Analysis of Algorithm 2

*Proof of Theorem 11.* The validity of Algorithm 2 relies on two things: first, it must correctly execute, meaning that a choice of $c_e$ must always exist; second, the set of elements it accepts must always be independent. The first is fairly straightforward: we have a choice of $2\alpha - 1$ colors, and are constrained to choose a distinct color from at most $\alpha - 1$ elements on the left and at most $\alpha - 1$ colors from the right. This constraint eliminates at most $2(\alpha - 1) < 2\alpha - 1$ colors, and so there is necessarily always a color available to choose.

Proving that the set of elements the algorithm accepts is always independent is equivalent to proving that the set of elements with each color is independent, which we will do via the three lemmas below.

**Lemma 7.** *The ordering $T$ maintained by the algorithm always satisfies the property that for any $j = 1, \ldots, m$, the elements in $S_j$ which have been presented appear contiguously in $T$.*

*Proof.* By induction (as the initial ordering is empty and so trivially satisfies the property), it suffices to prove that given an ordering $T$ satisfying the aforementioned property, inserting an element into $T$ in the manner done by Algorithm 2 will result in an ordering $T'$ which continues to satisfy the property. To show this, let $e, J$ be as defined in the algorithm. In the case that none of the elements of $J$ had appeared before (i.e. the "else" clause), appending $e$ to the end of $T$ ensures contiguity for all $j \in J$ and does not break the contiguity of other sets.

In the other case, the algorithm chooses the $j \in J$ that appeared before but did so the fewest times. In this case, $e$ is inserted after the last element in $T$ contained in $S_j$. This clearly ensures the contiguity of $S_j$ in $T$. Furthermore, for any other $l \in J$ such that $l$ had appeared before, we can argue by the laminar property of the sets in $F$ that all previously seen elements which were in $S_j$ are also in $S_l$. To see this, note that as $e$ is contained in both $S_j$ and $S_l$, it must be that either $S_j \subseteq S_l$ or $S_l \subseteq S_j$. In the former case, it is automatically true that all previously seen elements in $S_j$ are in $S_l$. In the latter case, if there was a previously seen element in $S_j$ that was not in $S_l$, then as all previously seen elements in $S_l$ are also in $S_j$, it would necessarily be true that the number of previously seen elements in $S_l$ was smaller than the number of previously seen elements in $S_j$, which contradicts the definition of $j$. We therefore have that all previously seen elements in $S_j$ are also in $S_l$, and so the algorithm, by inserting $e$ next to an element of $S_j$, also inserts it next to an element of $S_l$, ensuring the contiguity of $S_l$.

We must finally argue that there is no $l$ such that $S_l$ does not contain $e$ and the contiguity of $S_l$ is broken by the insertion of $e$ into $T$. To see this, note that if this were the case, then after the insertion of $e$ into $T$ to obtain $T'$, there would be some elements to the left of $e$ in $S_l$ and some elements to the right of $e$ in $S_l$. However, as $S_l$ was contiguous in $T$ prior to the insertion of $e$, it must have been that every element in between the leftmost such element and the rightmost such element was contained in $S_l$. In particular, letting $d$ be the element immediately to the left of $e$ and $f$ be the element immediately to the right of $e$ in $T'$, both $d, f$ must be contained in $S_l$. Recall that $e$ is inserted after the last element of $S_j$ in $T$; this implies that $d \in S_j$ and $f \notin S_j$. The fact that $d \in S_l, S_j$ means that one of $S_l, S_j$ must be a subset of the other. Meanwhile, the fact that $e \notin S_l$ implies that $S_j$ cannot be a subset of $S_l$, while the fact that $f \notin S_j$ implies that $S_l$ cannot be a subset of $S_j$. This is a contradiction, and so we are done. $\qquad\square$

**Lemma 8.** *The ordering $T$ and coloring $c$ maintained by the algorithm always satisfy the property that no $\leq \alpha$ consecutive elements in $T$ have the same color.*

*Proof.* We will again prove this inductively: we must show that given that $T$ satisfies the property, inserting an element $e$ into $T$ to obtain $T'$ and coloring it in the manner done by Algorithm 2 will maintain the property. To see this, we first note that for any set $A$ of $\leq \alpha$ consecutive elements in $T'$, the set $A \setminus \{e\}$ contains elements of distinct colors; this is because $A \setminus \{e\}$ then corresponds to a set of $\leq \alpha$ consecutive elements in $T$. We then note that for any such $A$ containing $e$, $e$ has a distinct color from the elements of $A \setminus \{e\}$ – this is because $e$ is specifically chosen to have a distinct color from the $\alpha - 1$ elements to its left and the $\alpha - 1$ elements to its right in $A$, and by the definition of $A$, if it contains $e$, it cannot contain any farther away elements. We therefore have that all $A$ consist of elements of distinct colors as desired. $\qquad\square$

**Lemma 9.** *Suppose that we have an ordering $T$ of the elements of the laminar matroid $M$ such that for any $j = 1, \ldots, m$, the elements in $S_j$ which have been presented appear contiguously in $T$, and a coloring of the elements of $M$ such that no $\leq \alpha$ consecutive elements in $T$ have the same color. Then for any color $c$, the set of elements with color $c$ is independent.*

*Proof.* It suffices to prove that for any color $c$ and for any $j = 1, \ldots, m$, at most $r_j$ elements in $S_j$ are colored $c$. To prove this, first note that $S_j$ must contain at most $\alpha r_j$ elements. This is because the entire matroid $M$ can be partitioned into $\alpha$ independent sets; each such set contains at most $r_j$ elements from $S_j$, and so in total there cannot be more than $\alpha r_j$ elements in $S_j$. This then means that $S_j$ is a contiguous subsequence of $T$ of length at most $\alpha r_j$.

We now note that for any contiguous subsequence of $T$ of length $l$, said subsequence can contain at most $\lceil \frac{l}{\alpha} \rceil$ elements colored $c$. We can see this by partitioning said subsequence into its first $\alpha$ elements, its next $\alpha$ elements, and so on; there are exactly $\lceil \frac{l}{\alpha} \rceil$ parts, and as each part consists of $\leq \alpha$ consecutive elements, it can contain at most one element of color $c$.

It follows that $S_j$, being a contiguous subsequence of $T$ of length at most $\alpha r_j$, contains at most $\lceil \frac{\alpha r_j}{\alpha} \rceil = r_j$ elements of color $c$ as desired. $\qquad\square$

We can now apply Lemma 7 and Lemma 8 to see that the final values of $T, c$ satisfy the preconditions of Lemma 9, which can then be applied to see that the set of elements with any particular color $c$ is independent, as desired.

It only remains to note that, as we color elements with $1, \ldots, 2\alpha - 1$ and select the color whose elements to accept uniformly at random from $1, \ldots, 2\alpha - 1$, each element is accepted with probability exactly $\frac{1}{2\alpha - 1}$, and so the algorithm is $(2\alpha - 1)$-fair as desired. $\qquad \square$

*Proof of Corollary 3.* By Theorem 11, Algorithm 2 is $(2\alpha - 1)$-fair, which implies that it is $2\alpha$-fair. We can then apply Theorem 8 with $h(M) = \alpha(M)$ and $c = 2$ to obtain the desired result. $\qquad \square$

## C.5 Sampling in the random order setting

**Definition 5.** *Define the* sampling model *for fair matroid selection to be as follows:*

- *An algorithm $A$ specifies $m$ probabilities $p_1, \ldots, p_m$ such that $p_1 + \cdots + p_m \leq 1$.*

- *A matroid $M = (E, \mathcal{I})$ is chosen, unknown to the algorithm.*

- *We define random sets $S_1, \ldots, S_m, T \subseteq E$ as follows. For each $j = 1, \ldots, m$ and each $e \in E$, $e$ is placed into exactly one of $S_1, \ldots, S_m, T$ – specifically, $e$ is placed into $S_j$ with probability $p_j$.*

- *$A$ observes the values of $S_1, \ldots, S_m$.*

- *$A$ is then presented the elements of $T$ in a uniformly random order, and must choose to accept or reject each online, subject to independence of the accepted elements.*

- *The fairness of $A$ is defined as usual – the inverse of the minimum probability $p_e$ with which an element $e \in E$ is accepted by $A$ (noting that $e$ not being placed in $T$ prevents it from being accepted and therefore decreases $p_e$).*

**Theorem 18.** *Let $C$ be a class of matroids which is closed under restriction (i.e. if a matroid $M = (E, \mathcal{I})$ is in $C$, then the matroid $M' = (E', \mathcal{I}')$ where $E' \subseteq E$ and $\mathcal{I}' = \mathcal{I} \cap 2^{E'}$ is also in $C$). If an $f$-fair algorithm exists for $C$ in the sampling model of fair matroid selection, then an $f$-fair algorithm exists for random order fair matroid selection on $C$.*

*Proof.* We will define an algorithm $A'$ for random order fair matroid selection which simulates the sampling of $S_1, \ldots, S_m$ as well as the behavior of $A$. The algorithm works as follows: we define random sets $R_1, \ldots, R_m, R_{m+1}$ by sampling the integers $1, \ldots, n$ in the same way that $S_1, \ldots, S_m, T$ sample the elements of $E$. Thus, for each $a = 1, \ldots, n$, $a$ is included in exactly one of $R_1, \ldots, R_{m+1}$, such that for each $j = 1, \ldots, m$, $a$ is included in $R_j$ with probability $p_j$ (and so $a$ is included in $R_{m+1}$ with probability $1 - (p_1 + \cdots + p_m)$). We then observe elements of $E$ as follows: we place the first $|R_1|$ elements into $S_1$, the next $|R_2|$ elements into $S_2$, and so on. Prior to the last $|R_{m+1}|$ elements, we present $S_1, \ldots, S_m$ to $A$, then allowing $A$ to act on those remaining $|R_{m+1}|$ elements, which are considered to be in $T$.

In order to prove the theorem, it suffices to show that the construction of $S_1, \ldots, S_m, T$ by $A'$ is identical to their definition by sampling in the theorem statement. To see this, first note that the behavior of $A'$ can alternatively be described as follows: we construct a sequence $r_1, \ldots, r_n$ where $r$ contains first the elements of $R_1$ in some order, then the elements of $R_2$ in some order, and so on. Then, let $e_1, \ldots, e_n$ be the elements of $E$ in the order they are presented. We define $\tau : E \to \{1, \ldots, n\}$ by $\tau(e_i) = r_i$.

Now note that because the order of elements $e_1, \ldots, e_n$ is chosen uniformly at random independent of the definition of $R_1, \ldots, R_m$, the mapping $\tau$ is also a uniformly random bijection, meaning that although its definition depends on the order of $r_1, \ldots, r_n$ and therefore on the sets $T_1, \ldots, T_m$, as a random variable it can be regarded as independent of $R_1, \ldots, R_m$.

We then note that $e \in S_j$ iff $\tau(e) \in R_j$. Furthermore, the decisions of whether to include $\tau(e) \in R_j$ for each $(e, j)$ are made randomly independently from each other and, as we have just shown, also independently from $\tau$, meaning that we can interpret $\tau$ as being defined first, after which we independently decide whether $\tau(e) \in R_j$ for each $(e, j)$. This is then equivalent to deciding whether $e \in S_j$ for each $(e, j)$ as desired. $\qquad \square$

## C.6 Relaxing knowledge of the arboricity

*Proof of Theorem 12.* We will specifically show that the $A'$ we define is $4f(2\alpha+1)$-fair; $4f(2\alpha+1)$ will then also be $O(f(\alpha))$ as $f$ is polynomial. We will define $A'$ in the setting of Theorem 18, which can then be applied to yield an algorithm in the usual setting with the same fairness. $A'$ will work as follows: we set $m=1$ and $p_1 = \frac{1}{2}$, meaning that we observe a single set $S_1 \subseteq E$ into which each element is sampled with probability $\frac{1}{2}$. We then compute the arboricity $\alpha'$ of $S_1$, and finally run $A_{2\alpha'+1}$ on the remaining elements.

To analyze $A'$, consider a specific element $e \in E$ – we will show that $e$ is accepted with probability at least $\frac{1}{4f(2\alpha+1)}$. First note that $e$ is sampled into $S_1$ with probability $\frac{1}{2}$, and therefore is *not* sampled into $S_1$ with probability $\frac{1}{2}$; we will condition on the latter event. Now note that $\alpha' \leq \alpha$, and so $2\alpha' + 1 \leq 2\alpha + 1$, meaning that the algorithm $A_{2\alpha'+1}$ we run will achieve a fairness of at most $f(2\alpha+1)$ on the elements $E \setminus S_1$ as long as $2\alpha' + 1$ is at least the arboricity of $E \setminus S_1$.

It thus only remains to show that $2\alpha' + 1$ is at least the arboricity of $E \setminus S_1$ with probability at least $\frac{1}{2}$. To show this, we will show the stronger statement that $2\alpha' + 1$ is at least $\alpha$ with probability at least $\frac{1}{2}$. First observe that the set $E \setminus \{e\}$ has arboricity at least $\alpha - 1$, as adding back $e$ can increase the number of independent sets needed to cover the set by at most 1. Now note that by Theorem 6, there exists a set $A \subseteq E \setminus \{e\}$ such that $\lceil \frac{|A|}{\text{rank } A} \rceil = \alpha(E \setminus \{e\}) \geq \alpha - 1$. With probability at least $\frac{1}{2}$, at least half of $A$ will be sampled into $S_1$, meaning that, defining $A' = A \cap S_1$, as $\text{rank } A' \leq \text{rank } A$, $\lceil \frac{|A'|}{\text{rank } A'} \rceil \geq \lceil \frac{|A'|}{\text{rank } A} \rceil \geq \frac{\alpha-1}{2}$. As $\alpha' \geq \lceil \frac{|A'|}{\text{rank } A'} \rceil$ by the easy direction of Theorem 6, we then have that with probability at least $\frac{1}{2}$, $2\alpha' \geq \alpha - 1$ and so $2\alpha' + 1 \geq \alpha$ as desired. $\qquad \square$

## C.7 Relaxing knowledge of the rank

We next relax the knowledge of the rank in the random order setting using the same proof as Theorem 12.

**Theorem 19.** *Suppose that for some class $C$ of matroids closed under restriction, we are given an algorithm $A_k$ parameterized by the rank $k$ and a nondecreasing $f : \mathbb{Z}_+ \to \mathbb{R}$ that grows polynomially in $k$, such that the algorithm $A_k$ is $f(k)$-fair when the input matroid $M \in C$ has rank $k$. Then there exists an algorithm $A'$ for random order fair matroid selection without knowledge of $k$ that is $O(f(k))$-fair on $M \in C$.*

*Proof.* We will establish that the $A'$ we construct ensures $4f(2k+1)$-fairness. Since $f$ is a polynomial function, we have $4f(2k+1) \leq O(f(k))$.

The construction of $A'$ takes place within the framework of Theorem 18. From there, we can extend the result to obtain an algorithm in the standard setting while preserving the same fairness guarantee. The procedure for $A'$ is as follows: we set $m=1$ and define $p_1 = \frac{1}{2}$, meaning that we sample a subset $S_1 \subseteq E$, where each element is included independently with probability $\frac{1}{2}$. We then determine the rank $k'$ of $S_1$ and subsequently apply $A_{2k'+1}$ to the remaining elements.

To analyze $A'$, consider an arbitrary element $e \in E$. Our goal is to show that $e$ is selected with probability at least $\frac{1}{4f(2k+1)}$. Condition on the event that $e \notin S_1$, which happens with probability $\frac{1}{2}$. Since $k' \leq k$, it follows that $k' + 1 \leq 2k + 1$. Therefore, the algorithm $A_{2k'+1}$ we execute will maintain fairness at most $f(2k+1)$ on the remaining elements $E \setminus S_1$, provided that $2k' + 1$ meets or exceeds the rank of $E \setminus S_1$.

The only remaining step is to establish that, with probability at least $\frac{1}{2}$, the rank of $E \setminus S_1$ does not exceed $2k' + 1$. We will prove the stronger claim that this quantity is at least $k$ with probability at least $\frac{1}{2}$. Observe first that the set $E \setminus e$ has rank at least $k - 1$. This implies that there exists some subset $A \subseteq E \setminus e$ that is independent and has $|A| \geq k - 1$. With probability at least $\frac{1}{2}$, at least half of the elements in $A$ appear in $S_1$, ensuring that $k' \geq \frac{k-1}{2}$. Rearranging this inequality, we obtain $k \geq 2k' + 1$, completing the proof. $\qquad \square$

## C.8 Properties of the $d$-level

*Proof of Theorem 14.* We first show that if $l_d$ is $\infty$ on some vertices, then $G$ is not $d$-degenerate. Let $S \subseteq V$ be the set of vertices on which $l_d$ is $\infty$. Then all vertices in the subgraph defined by $S$ have degree greater than $d$. It follows that no ordering of $V$ satisfying the definition of $d$-degeneracy can exist: letting $v$ be the element of $S$ which appears latest in the ordering, it would have more than $d$ edges to the vertices in $S \setminus \{d\}$, all of which appear earlier in the ordering.

Conversely, if $l_d$ is never $\infty$, that $G$ is $d$-degenerate follows from the second part of the theorem, that we can order $V$ in decreasing order by $l_d$ to obtain an ordering satisfying the definition of $d$-degeneracy. To see that such an ordering is valid, we must show that any vertex $v$ has at most $d$ edges to vertices $u$ such that $l_d(u) \geq l_d(v)$. This follows from the fact that $v$ can only be removed in iteration $j$ (in the definition of $d$-level), and therefore be assigned $l_d(v) = j$, if at that points its degree is at most $d$. The remaining vertices at that point are precisely the vertices $u$ that will be assigned $l_d(u) \geq j$, completing the proof.

To see the final part of the theorem, we simply note that the process defining the $d$-level exactly corresponds to the computation of a function with the given properties. Specifically, we first assign $l_d(v) = 0$ to all vertices with degree at most $d$ as described. Then, any other vertex $v$ is assigned $l_d(v) = j$ iff iteration $j$ is the precise iteration in which the degree of $v$ became at most $d$. It follows that, as vertices are removed in order of nondecreasing $l_d$, the neighbor $u$ of $v$ with the $d+1$-th highest value of $l_d$ was removed in iteration $j-1$, meaning that $l_d(u) = j - 1$ and so $l_d(v) = l_d(u) + 1$ as desired. $\square$

## C.9 Analysis of Algorithm 3

*Proof of Theorem 15.* We first note that Algorithm 3 is valid – the set of elements it accepts will always be independent. This follows from the fact that it orders the vertices of $G$, and accepts at most one edge from each vertex to an earlier vertex in the ordering. It thus remains only to analyze the fairness of the algorithm.

We will do this as described before, by fixing a specific vertex $v$ and analyzing the sampled edges outgoing from $v$ after having already chosen the sampled edges outgoing from other vertices. The first tool we need for this is a modification of the notion of the $d$-level that does not depend on the edges outgoing from $v$.

**Definition 6.** *The $v$-fixed $d$-level, $l_d^v : V \to \mathbb{N}_0 \cup \{\infty\}$, is defined on the vertices of a directed graph $G$ by following the same iterative process as was used to define $l_d$, except that vertex $v$ is never removed. Thus, we perform iterations numbered starting from $0$, such that in iteration $j$ we remove all vertices $u \neq v$ from $G$ such that $u$ has outdegree at most $d$, and set $l_d^v(u) = j$; we terminate once all remaining vertices other than $v$ have outdegree greater than $d$, at which point we set $l_d^v(u) = \infty$ for all such $u$ (which necessarily includes $v$ itself).*

Note that while the $d$-level on $G$, and therefore $H$ (as a subgraph of $G$) is guaranteed to be finite for all vertices, the $v$-fixed $d$-level on $H$ is not, and in particular will be infinite for $v$ itself.

The key utility of the $v$-fixed $d$-level is that it actually serves as an almost exact proxy for the $d$-level for our purposes. Specifically, for all vertices with $d$-level less than the $d$-level of $v$, their $d$-level is identical to their $v$-fixed $d$-level. This is expressed in the below lemma.

**Lemma 10.** *On a directed graph $G$, let $x = l_d(v)$. Then for all $y < x$, and for all vertices $u$ in $G$, $l_d(u) = y$ iff $l_d^v(u) = y$.*

*Proof.* The key insight of this lemma is simply that the iterative processes used to define each of $l_d$ and $l_d^v$ are identical up to (but not including) iteration $x$: in the former process, we know that $v$ is necessarily not at an iteration prior to iteration $x$ by the definition of $l_d$, while in the latter process, we explicitly constrain $v$ to never be removed. Therefore, all iterations prior to iteration $x$ are identical in both processes, and so it follows that the set of vertices removed at iteration $y < x$ in one process is identical to the set of vertices removed at iteration $y$ in the other. $\square$

We can then apply this lemma to show, roughly, that if enough outgoing edges from $v$ are sampled, then the total number of edges in $G$ between $v$ and vertices with $d$-level (defined based on $H$) at least $l_d(v)$ is limited.

**Lemma 11.** *Let $G$ be the input graph to the algorithm, let $H$ be as defined by the algorithm, let $v$ be a vertex in $G$, and let $l_d, l_d^v$ be defined on $H$ (meaning that $l_d$ is as computed by the algorithm). Sort the neighbors of $v$ in decreasing order of $l_d^v$ to obtain a sequence $s$, breaking ties by label. If for some $b$, there are at least $d + 1$ neighbors $u$ among $s_1, \ldots, s_b$ such that the edge $v \to u$ is included in $H$, then the only neighbors $w$ of $v$ that can satisfy $l_d(w) \geq l_d(u)$ are those in $s_1, \ldots, s_{b-1}$.*

*Proof.* Let $x = l_d(v)$. We first claim that $x > l_d(s_b)$. Suppose to the contrary that $x \leq l_d(s_b)$. Then we can see by Lemma 10 that the set of vertices $u$ satisfying $l_d(u) < x$ is precisely the set of vertices satisfying $l_d^v(u) < x$. Therefore, all neighbors $u$ of $v$ satisfying $l_d^v(u) \geq x$ satisfy $l_d(u) \geq x$. This then includes all of $s_1, \ldots, s_b$, as $l_d^v(s_b) = x$, and $s$ is sorted in decreasing order of $l_d^v$. We then have that $v$ has at least $d + 1$ neighbors $u$ with $l_d(u) \geq x = l_d(v)$. This contradicts the clause of Theorem 14 stating that $l_d(v)$ is exactly 1 more than the $d + 1$-th highest $l_d(u)$ among neighbors $u$ of $v$. We have thus shown that $x > l_d(s_b)$.

It now remains to show that the neighbors of $v$ other than $s_1, \ldots, s_{b-1}$, meaning the neighbors $u$ among $s_b, \ldots, s_{\deg(v)}$, cannot satisfy $l_d(u) \geq x$. We first note that by the definition of $s$, such $u$ satisfy $l_d^v(u) \leq l_d^v(s_b) < x$. We can then apply Lemma 10 to see that such $u$ satisfy $l_d(u) = l_d^v(u) < x$. $\square$

The above lemma is essential – it allows us to work on an ordering of $v$'s neighbors that is independent of the sampled outgoing edges from $v$, meaning that we can view those edges (more precisely, the subset that was not already sampled in the other direction) as being sampled *after* the ordering is fixed in order to bound the number of neighbors $u$ of $v$ satisfying $l_d(u) \geq l_d(v)$, which can now be done using techniques from concentration bounds. Specifically, we will apply a Chernoff bound to show that the number of edges from $v$ to vertices earlier in Algorithm 3's ordering is close to $d$ with high probability.

We first note that an edge $(u, v)$ is sampled as $u \to v$ with probability $\frac{1-q}{2}$, as $v \to u$ with probability $\frac{1-q}{2}$, and not sampled with probability $q$; therefore, given that it was not sampled as $v \to u$, the probability that it is not sampled at all is $x = \frac{q}{q + \frac{1-q}{2}} = \frac{2q}{1+q}$. We additionally define $\delta = d^{-\frac{1}{8}}$, $b = \lceil \frac{d}{1 - x(1+\delta)} \rceil$, and $P = e^{-\frac{xb\delta^2}{3}}$. We now prove the following lemma, which relies on Lemma 11.

**Lemma 12.** *First note that for $d \geq 3$ we have $b > d$. Now, let $v$ be a vertex in $G$, and let $e$ be an edge in $G$. Define $Y$ to be the number of edges that are not sampled into $H$, not including $e$, between $v$ and a vertex $u$ such that $l_d(u) \geq l_d(v)$. Conditioned on $e$ not being sampled into $H$, the probability that $Y \geq b - d$ is at most $P$.*

*Proof.* We must first prove that for $d \geq 3$, we have $b > d$. Recall that $b = \lceil \frac{d}{1 - x(1+\delta)} \rceil$, where $x = \frac{2q}{1+q}$, $q = d^{-\frac{1}{4}}$, and $\delta = d^{-\frac{1}{8}}$; it suffices to show that $x(1 + \delta) < 1$. This then expands as $\frac{2q}{1+q}(1 + \delta) < 1$, which can be rewritten as $2q(1 + \delta) < 1 + q$. Substituting in $q$ and $\delta$, this is $\frac{2}{3}d^{-\frac{1}{4}}(1 + d^{-\frac{1}{8}}) < 1 + \frac{1}{3}d^{-\frac{1}{4}}$, which we can multiply by $3d^{\frac{3}{8}}$ to attain $2(d^{\frac{1}{8}} + 1) < 3d^{\frac{3}{8}} + d^{\frac{1}{8}}$, i.e. $3d^{\frac{3}{8}} - d^{\frac{1}{8}} - 1 > 0$. Letting $u = d^{\frac{1}{8}}$, this is equivalent to showing that $3u^3 - u - 1 > 0$ for $u \geq 3^{\frac{1}{8}}$. As $u^3$ increases faster than $u$ for $u \geq 1$, $3u^3 - u - 1$ is increasing for $u \geq 1$; furthermore, $3u^3 - u - 1$ is equal to 1 when $u = 1$. Therefore, $3u^3 - u - 1$ is positive for $u \geq 1$, and so in particular is positive for $u \geq 3^{\frac{1}{8}}$ as desired.

Proceeding, for convenience, we will first prove the following sublemma, encapsulating our specific use of the Chernoff bound.

**Lemma 13.** *Let $X_1, \ldots, X_b$ be independent random variables taking values in $\{0, 1\}$, such that each $X_j$ is 1 with probability $x$. Let $X = X_1 + \cdots + X_b$ be their sum; then,*

$$\Pr[X \geq xb(1 + \delta)] \leq e^{-\frac{xb\delta^2}{3}}.$$

*Proof.* We note that a well-known Chernoff bound for the upper tail states that for a random variable $X$ with mean $\mu$ equal to the sum of independent random variables taking values in $\{0, 1\}$, we have $\Pr[X \geq (1+\delta)\mu] \leq e^{-\frac{\mu\delta^2}{3}}$ for $0 < \delta < 1$. In our case, we have $\mu = xb$, from which the conclusion immediately follows. $\qquad\square$

Now note using the definition of $b$ that for any $X$, $b - X \leq d$ implies $X \geq xb(1 + \delta)$, which we can see using the contrapositive as follows: if $X < xb(1 + \delta)$, then $b - X > b - xb(1 + \delta) = (1 - x(1 + \delta))b \geq d$, meaning that $b - X > d$ as needed. This is useful, as it means that $\Pr[b - X \leq d] \leq \Pr[X \geq xb(1 + \delta)]$ and so is also bounded by $P$. We now apply this to prove the remainder of the lemma, which we restate as the below sublemma:

**Lemma 14.** *Let $v$ be a vertex in $G$, and let $e$ be an edge in $G$. Define $Y$ to be the number of edges that are not sampled into $H$, not including $e$, between $v$ and a vertex $u$ such that $l_d(u) \geq l_d(v)$. Conditioned on $e$ not being sampled into $H$, the probability that $Y \geq b - d$ is at most $P$.*

*Proof.* We will show that this holds after fixing whether each edge not outgoing from $v$ is sampled into $H$; the lemma itself then follows by averaging over all such cases. Now define $s$ to be the neighbors of $v$ in decreasing order of $l_d^v$, and define $b'$ to be the smallest integer such that among $s_1, \ldots, s_{b'}$, there are exactly $b$ neighbors $u$ such that the edge $(v, u)$ is not $e$ and was not sampled as $u \to v$. Note that such a $b'$ may not exist; we therefore proceed with case analysis based on whether $b'$ exists or not.

Given that $b'$ exists, let $u_1, \ldots, u_b$ be the $b$ neighbors among $s_1, \ldots, s_{b'}$ such that for all $j$, $(v, u_j) \neq e$ and $u_j \to v$ was not sampled into $H$. Now define $X_1, \ldots, X_b$ so that $X_j$ is 0 if $v \to u_j$ is sampled into $H$, and 1 otherwise. It follows that $X_1, \ldots, X_b$ are independent variables taking values in $\{0, 1\}$, each with mean $x$ (following from the definition of $x$). We can now apply Lemma 13 along with the note that followed it to see that the probability that $b - X \leq d$ is at most $P$; thus, with probability at least $1 - P$, we have that $b - X > d$. The quantity $b - X$ is equal to the number of $j$ such that $v \to u_j$ is sampled; thus, when $b - X > d$, we have that at least $d + 1$ of the edges from $u$ to $u_1, \ldots, u_b$ were sampled, and therefore at least $d + 1$ of the edges from $u$ to $s_1, \ldots, s_{b'}$ were sampled.

We can now apply Lemma 11 to see that with probability at least $1 - P$, the only neighbors $u$ of $v$ that can satisfy $l_d(u) \geq l_d(v)$ are those among $s_1, \ldots, s_{b'}$. From these, only the neighbors $u_1, \ldots, u_b$ satisfy the condition that the edge connecting them is both not $e$ and was not sampled in the direction incoming to $v$, and of those $b$ neighbors, at least $d + 1$ had their edges to $v$ sampled outgoing from $v$, meaning that only the remaining $b - d - 1$ neighbors contribute to $Y$. We therefore have that with probability at least $1 - P$, $Y \leq b - d - 1$; the desired conclusion follows.

In the case that no $b'$ exists, we instead consider all neighbors of $v$, and define $m$ to be the number of these numbers who are connected to $v$ by an edge other than $e$ that was not sampled incoming to $v$; we then let these neighbors be $u_1, \ldots, u_m$. We again define random variables $X_1, \ldots, X_m$ such that $X_j$ is 0 if $v \to u_j$ is sampled into $H$, and 1 otherwise. In the case that $m = b$, we had by Lemma 13 that $\Pr[X \geq xb(1 + \delta)] \leq P$; this probability cannot increase when we have less variables with the same mean, so we necessarily have here as well that $\Pr[X \geq xb(1 + \delta)] \leq P$. We can then apply the same note to see that $\Pr[b - X \leq d] \leq P$. We thus have like before that only the neighbors $u_1, \ldots, u_m$ satisfy the condition that the edge connecting them is both not $e$ and was not sampled in the direction incoming to $v$, and of those $m < b$ neighbors, with probability at least $1 - P$, $d + 1$ had their edges to $v$ sampled outgoing from $v$, meaning that only the remaining $m - d - 1 < b - d - 1$ neighbors contribute to $Y$. We therefore have that with probability at least $1 - P$, $Y \leq m - d - 1 < b - d - 1$; the desired conclusion again follows. $\qquad\square$

$\qquad\square$

We can now move to the final key lemma, lower bounding the probability that a particular edge $e$ is taken. We prove this lemma by applying the union bound to see that with high probability, both of its endpoints satisfy the above lemma, and so regardless of which is the later endpoint in the ordering, $e$ is not competing with significantly more than $d$ edges in order to be accepted.

**Lemma 15.** *For any edge $e$ in the graph $G$, Algorithm 3 accepts $e$ with probability at least $(1 - 2P)\frac{q}{b-d}$.*

*Proof.* Let $e = (v_1, v_2)$. We first condition on the event that $e$ is not sampled into either $S_1$ or $S_2$, which occurs with probability $q$. We can then apply Lemma 12 to see that for each $v = v_1, v_2$, the probability that, conditioned on $e$ not being sampled, there are more than $b - d - 1$ other edges not sampled that connect $v$ to a vertex $u$ with the same or higher value of $l_d$, is at most $P$. Thus, by the union bound, this is true for either $v_1$ or $v_2$ with probability at most $2P$. Therefore, with probability at least $1 - 2P$, for $v$ equal to both $v_1, v_2$ we have the property that there are at most $b - d - 1$ other edges with later endpoint $v$ that could be accepted. In particular, this holds true for the vertex $v \in \{v_1, v_2\}$ which is the later endpoint of $e$.

We then have that there are at most $b - d$ edges with later endpoint equal to this $v$ which could be accepted, and as the unsampled edges are presented in random order, $e$ is the first such edge, and so accepted by the algorithm, with probability at least $\frac{1}{b-d}$. Combined with the $q$ probability of $e$ not being sampled and the $1 - 2P$ probability that both endpoints satisfy the previously described property, we get that $e$ is taken with probability at least $q \cdot (1 - P) \cdot \frac{1}{b-d}$ as desired. $\qquad\square$

The following last lemma is purely technical and serves to extract the asymptotic behavior of the concrete values we have chosen for $b, q, P$.

**Lemma 16.** $(1 - 2P)\frac{q}{b-d}$ *is positive for* $d \geq 3$, *and its reciprocal is* $(2 + o(1))d$.

*Proof.* We first prove that $(1 - 2P)\frac{q}{b-d}$ is positive when $d \geq 3$. Recall again that $b = \lceil \frac{d}{1-x(1+\delta)} \rceil$, $x = \frac{2q}{1+q}$, $q = d^{-\frac{1}{4}}$, $\delta = d^{-\frac{1}{8}}$, and $P = e^{-\frac{xb\delta^2}{3}}$. We have that $q$ itself is positive, and we have already shown in Lemma 12 that $b > d$, meaning that the term $\frac{q}{b-d}$ is positive. We therefore need only show that $P < \frac{1}{2}$, from which it follows that $1 - 2P$ is positive. This is equivalent to $\frac{xb\delta^2}{3} > \ln 2$. We first note that by definition, $b \geq \frac{d}{1-x(1+\delta)}$, meaning that it suffices to show that $\frac{x\delta^2}{3} \cdot \frac{d}{1-x(1+\delta)} > \ln 2$. Also note that $\ln 2 < 1$, and so it similarly suffices to show that $\frac{x\delta^2}{3} \cdot \frac{d}{1-x(1+\delta)} > 1$. We now first rewrite this as $x\delta^2 d > 3(1-x(1+\delta))$, then substitute in $x = \frac{2q}{1+q}$ to get $\frac{2q}{1+q}\delta^2 d > 3(1 - \frac{2q}{1+q}(1+\delta))$, which can be multiplied by $1 + q$ to get the equivalent inequality $2q\delta^2 d > 3((1+q) - 2q(1+\delta))$.

We now recall that $q = \frac{1}{3}d^{-\frac{1}{4}}$ and $\delta = d^{-\frac{1}{8}}$; if we let $u = d^{\frac{1}{8}}$, then these are $q = \frac{1}{3u^2}$ and $\delta = \frac{1}{u}$ (as well as $d = u^8$), meaning that we would like to show that $2 \cdot \frac{1}{3u^2} \cdot \frac{1}{u^2} \cdot u^8 > 3((1+\frac{1}{3u^2}) - \frac{2}{3u^2}(1+\frac{1}{u}))$ for $u \geq 3^{\frac{1}{8}}$. This simplifies to $\frac{2}{3}u^4 > 3(1 - \frac{1}{3u^2} - \frac{2}{3u^3})$. We can then multiply by $3u^3$ to get the equivalent inequality $2u^7 > 3(3u^3 - u - 2)$, which can be rearranged as $2u^7 - 6u^3 + 3u + 6 > 0$. We therefore must show that $2u^7 - 6u^3 + 3u + 6 > 0$ for $u \geq 3^{\frac{1}{8}}$; it then suffices to show that $2u^7 - 6u^3 + 6 > 0$ for nonnegative $u$, as the term $3u$ that we subtracted is also nonnegative. We can see this by noting that as $2u^7 - 6u^3 + 6$ goes to $+\infty$ as $u$ goes to $+\infty$, it is minimized at either a boundary (i.e. $u = 0$) or a critical point, meaning that it suffices to show that it is positive at those places. When $u = 0$, it simply takes the value 6. Then, to find critical points, we set its derivative to 0: thus, $0 = \frac{d}{du}(2u^7 - 6u^3 + 6) = 14u^6 - 18u^2 = 2u^2(7u^4 - 9)$. Thus, the critical points are $u = 0$ and $u = \pm(\frac{9}{7})^{\frac{1}{4}}$. We have already discussed $u = 0$, and we are only concerned with nonnegative $u$, meaning that we need only check the values of $2u^7 - 6u^3 + 6$ at $u = (\frac{9}{7})^{\frac{1}{4}}$. There, we see that $2u^7 - 6u^3 + 6 = 6 + \frac{1}{u}(2u^8 - 6u^4) = 6 + \frac{1}{u}[2(\frac{9}{7})^2 - 6(\frac{9}{7})] = 6 + \frac{2(9)^2 - 6(9 \cdot 7)}{7^2} \cdot \frac{1}{u} = 6 - \frac{216}{49} \cdot \frac{1}{u}$; because $u > 1$, we have that $6 - \frac{216}{49} \cdot \frac{1}{u} > 6 - \frac{216}{49} = \frac{6(49) - 216}{49} = \frac{78}{49} > 0$ as desired.

It remains to show that $\frac{b-d}{q} \cdot \frac{1}{1-2P}$ is $(2 + o(1))d$. We first recall that $b = \lceil \frac{d}{1-x(1+\delta)} \rceil$, and note that all of $q, x, \delta$ go to 0 as $d$ grows. It follows that $b = d(1 + x(1+\delta) + o(x(1+\delta))) = d(1 + x + o(x))$. We thus have that $b - d = d(x + o(x))$. We now recall that $x = \frac{2q}{1+q} = \leq 2q$, meaning that $\frac{b-d}{q} = \frac{d(2q+o(q))}{q} = (2 + o(1))d$. We finally write out $P$ as $e^{-\frac{xb\delta^2}{3}}$; as $x = \Theta(q) = \Theta(d^{-\frac{1}{4}})$, $b = \Theta(d)$, and $\delta = d^{-\frac{1}{8}}$, we have that $xb\delta^2 = \Theta(d^{-\frac{1}{4}+1-2\cdot\frac{1}{8}}) = \Theta(\sqrt{d})$; it follows that $P$ also goes to 0 as $d$ grows, meaning that $\frac{1}{1-2P} = 1 + o(1)$. We can therefore conclude that $\frac{b-d}{q} \cdot \frac{1}{1-2P} = (2 + o(1))d \cdot (1 + o(1)) = (2 + o(1))d$ as desired. $\qquad\square$

It follows from combining the results of Lemma 15 and Lemma 16 that Algorithm 3 is $(2+o(1))d$-fair for $d \geq 3$. □

*Proof of Corollary 4.* Given knowledge of $\alpha$, by Theorem 13 we note that letting $d = 2\alpha - 1$ (note that for $\alpha \geq 2$, $d \geq 3$ as needed; if $\alpha = 1$, the algorithm can simply accept all elements), the input graph must be $d$-degenerate. Thus, by Theorem 15, we can apply Algorithm 3, which will be $(2 + o(1))d = (4 + o(1))\alpha$-fair. We can then apply Theorem 12 to *this* algorithm to obtain an oblivious algorithm which is $O(\alpha)$-fair. □

### C.10 Adversarial order fair matroid selection for uniform matroids

In this section, we provide a short proof for the existence of an $\alpha$-fair algorithm for the $k$-uniform matroid in the adversarial order model. Assuming we have $n$ elements, it is clear that the arboricity is $\alpha = \lceil n/k \rceil$. We color the elements with $\alpha$ colors such that each $\alpha$ consecutive arriving elements have a different color. This can easily be achieved by coloring the element at time $i$ with the remainder of $i$ divided by $\alpha$. We then pick all of the elements corresponding to a single color, chosen uniformly at random.

Since $\alpha \geq n/k$, there are at most $k$ elements from any given color. Therefore, the final set has size at most $k$ and is therefore feasible. Additionally, since we have $\alpha$ colors, each element is accepted with probability $1/\alpha$, finishing the proof.

