# OpenReview forum: "Fair Matroid Selection"
_NeurIPS.cc/2025/Conference — NeurIPS 2025 poster_

### Official Review · Reviewer_YJe1 · 2025-07-01

**Clarity:** 2
**Significance:** 2
**Originality:** 2
**Rating:** 4
**Confidence:** 3

**Summary:**

The paper studies the fair matroid selection problem. This is studied in an online setting where elements of an unknown matroid arrive over time and one has to irrevocable accept/reject each element constrained on the final selection belongs to the independent set of the underlying matroid. As opposed to more classic online matroid selection problems, such as the matroid secretary problem, there is no objective function to optimize over. Instead, the paper says that an algorithm is $f$-fair if every element is selected with probability at least $1/f$, and the goal is to design algorithms to minimize $f$. The paper provides algorithms for both adversarial and random arrival models, analyzing their fairness guarantees in terms of matroid arboricity $\alpha$ and rank $k$. Special cases of laminar and graphic matroids were also considered.

**Questions:**

Corollary 1: Is logarithmic loss the best possible feasible function? Can you give a proof?

Line 238: What do you mean by "iteration"? Algorithm 1 has no notion of iteration besides elements arriving

Line 264: May I know which is the high school with this system and where did you find out the description given in the paragraph above? I am genuinely curious to learn about the setting.

Proof idea of Theorem 11 and Corollary 3: If elements in the same independent set are contiguous in the list you are maintaining, but then are given different colors, how are you ensuring that the set of all accepted elements form an independent set in the matroid?

Why is there no probability terms or expectations in Theorem 12? Since the arrival is random order and the approach is to perform some estimation on the first half of the arrivals, I was expecting some kind of randomized analysis to occur.

I think I am not properly understanding your definition of matchoid in the first paragraph of Appendix B. If $p > m$ and any element belongs in $\leq p$ (line 616), then $A = \emptyset$? Please clarify this confusion. Thanks.

Some writing issues / typos that I found:
- There are some missing definitions whose addition in Section 1 would improve the writing quality: laminar, graphic, matchoid
- Lines 80-84: The use of semicolon parses weirdly
- Theorem 8 and Corollary 1: You used $k$ for matroid rank, so maybe use another letter like $\ell$ here instead?
- Algorithm 1 renders badly. It should be shifted to the end of the paragraph. You can say "Algorithm 1 ... expressed in Theorem 10" instead of inserting an algorithm environment in the middle of a paragraph.
- Line 270: "learn the set" should be "*all* indices $j$ such that $e \in S_j$" or rewrite to say "minimal sized independent set that contains $e$" and then modify Algorithm 2 to *not* mention $J$ since it isn't used anywhere else anyway
- Line 274: "continuous" should be "continues"
- Algorithm 2 for loop: "the indices of sets" should be "*all* indices of sets"
- Algorithm 2: The statement "Let $N$ be the union ..." is not very precise. Could you give a simple explicit example? Adding such an example in the main paper would also help the presentation of the algorithm.
- Algorithm 2: From the statement "Choose $c_e$ to be any color...", it is unclear whether one should set $c_e = r$ when possible.
- I'm not sure what's the point of the exposition from Line 298 to 303
- In Section 4, the definition of $p_1, \ldots, p_m$ and $S_1, \ldots, S_m$ are kind of unnecessarily complicated. In the proof in the appendix, you have only used $m = 1$ and $p_1 = 1/2$. The writing could be much simplified to say that the idea is to use the first half of the arrivals to estimate $\alpha$. Also, consider changing "less than 1" to "strictly less than 1" to be more precise on Line 312
- Line 912: $r$ is not defined, but I suppose it refers to $r_1, \ldots, r_n$ on Line 911?
- Line 914: What is $t_i$?

**Ethical Concerns:**

["NO or VERY MINOR ethics concerns only"]

**Final Justification:**

After the authors clarified my doubts, I believe there is some amount of technical novelty and contribution of the work. However*, I feel that just optimizing $f$-fairness without an additional objective severely limits the practicality of the work (and the rebuttal did not convince me otherwise). Furthermore, I believe that the writing can be improved.

* I acknowledge that the sentences that follow are very subjective.

**Quality:**

2

**Strengths And Weaknesses:**

# Strengths
- The paper presents nearly tight characterizations for $f$-fair algorithms for the setting studied
- There was an algorithmic overview written in the main prose to explain the intuition behind each result
# Weaknesses
- From my understanding (I could be wrong), *all* elements of the matroid will arrive (e.g. the algorithmic idea behind Appendix C.6 is to observe first half of the arrivals to estimate $\alpha$). In that case, why is it not possible to first query all subsets to learn the independent set $\mathcal{I}$ of the matroid, flip a suitably biased coin to decide which subset $S \in \mathcal{I}$ of elements to pick (e.g. choose each of the $\alpha$ partitions with uniform probability), then just patiently wait and only accept elements in $S$? If what I described is true, then I do not see the point of this paper (though I believe I am wrong due to Line 176?).
- There is inadequate comparison with the broader fairness literature and a lack of justification for why it is sufficient to consider $f$-fair algorithms without any objective function to optimize over simultaneously
- Some parts of the writing can be further improved (see questions section)
- There are no experiments. However, I am viewing this work as a theoretical submission, so I believe that lack of experiments is not a complete showstopper if the theory is good

I am happy to revise my score upwards if the weaknesses and my questions mentioned are adequately addressed.

---

> ### Author Rebuttal · Authors · 2025-07-30
>
> We thank reviewer YJe1 for your thorough feedback.
>
> > From my understanding (I could be wrong), all elements of the matroid will arrive…
>
> We note that although all elements of the matroid will arrive, the matroid is not known ahead of time. Thus, it is not possible to make independence queries on all subsets, as none of the elements of the matroid are known; it is only possible to make independence queries on subsets containing only elements which have already arrived. It is therefore not possible to partition the matroid ahead of time as you are suggesting in the online setting, necessitating the nontrivial algorithms we present in our paper. We further note that the algorithm you suggest is in fact the exact algorithm we use to prove our Theorem 7, showing that in an offline setting you can indeed achieve $\alpha$-fairness.
>
> > …lack of justification for why it is sufficient to consider $f$-fair algorithms…
>
> We essentially view the fairness parameter $f$ to be the parameter we are trying to optimize (with lower parameters denoting more fairer algorithms). While we agree that it is interesting to simultaneously optimize over an additional objective, our work shows that even without such considerations the problem is already highly non-trivial. We agree that this is a good direction for future work however and will add a brief discussion in the paper.
>
> > Corollary 1: Is logarithmic loss the best possible feasible function? Can you give a proof?
>
> We note that logarithmic loss is indeed the best possible feasible function in the sense that any feasible function has at least logarithmic loss. This is due to the fact that (in combination with Theorem 9) the function $f(x) = O(x \cdot \log x)$ has a sum of inverses which diverges. Thus, any function $g$ whose sum of inverses converges must grow strictly faster than $O(x \cdot \ln x)$ and thus must have at least logarithmic loss. The specific function $f(x) = O(x \cdot (log x)^2)$ that we use does not have the asymptotically smallest loss; for example, you could consider the function $f(x) = O(x \cdot \log x \cdot (\log \log x)^2)$, which has asymptotically smaller loss (which you pay for in the constant factor), but still logarithmic loss.
>
> > What do you mean by "iteration"?...
>
> By "iteration" we were referring to iterations of the previously stated argument, which colored $\frac 1 \alpha$ of the remaining (i.e. not accounted for by previous colors) elements, rather than iterations in the algorithm itself.
>
> > May I know which is the high school with this system…
>
> The high school in question is Thomas Jefferson High School for Science and Technology, commonly abbreviated as TJHSST. While we are unable to provide external links in our rebuttal, we suggest that searching the terms "TJHSST" and "geographic lottery" or "admissions changes" online may yield relevant information.
>
> > Proof idea of Theorem 11 and Corollary 3…
>
> We clarify that the sets $S_j$ are not independent sets, but rather the sets in the laminar family that defines the matroid, via the restriction that an independent set can have at most $r_j$ elements from the set $S_j$. Thus, giving elements in the same set $S_j$ different colors helps to maintain an independent set.
>
> > Why is there no probability terms or expectations in Theorem 12?
>
> We note that the definition of $f$-fairness already involves a probability term, as it is defined as the minimum probability of acceptance over all elements; the respective probabilities of acceptance are taken not only over the randomness of the algorithm but also the randomness of the order.
>
> > I think I am not properly understanding your definition of matchoid in the first paragraph of Appendix B…
>
> We first note that in a matchoid, we would generally expect to have $p \leq m$. If $p > m$, then there is essentially no constraints on the number of matroids each element belongs to (since any element can belong to at most $m$ matroids anyway). In this case, we do not necessarily have $A=\emptyset$.
>
> > Some writing issues / typos that I found:
>
> We thank the reviewer for finding and detailing these writing issues and will be sure to fix them in the next revision of the paper.
>
> If you feel that our response has adequately addressed your major concerns, we would appreciate it if you could possibly adjust your score accordingly.

---

> > ### Comment · Reviewer_YJe1 · 2025-08-02
> >
> > Thank you for your responses. You have clarified my doubts and I have no further questions, and have adjusted my score accordingly. Below are some additional follow-up remarks to your rebuttal.
> >
> > ### Weakness 1
> >
> > I see! This setting makes a lot more sense. Thank you for the clarification. Please make it more explicit that even though all elements arrive, the independence queries can only be performed elements that have already arrived. I must have missed it somewhere...
> >
> > ### Justification for $f$-fair without additional objectives
> >
> > I agree that what you investigated is a valid mathematical problem. I am hoping to see some practical justifications for why it is an worthy problem setting. If it is just purely for mathematical interest, that is also fine (but be upfront)...
> >
> > ### TJHSST
> >
> > Very interesting. Thank you for sharing.

---

### Official Review · Reviewer_eEjL · 2025-07-01

**Clarity:** 4
**Significance:** 2
**Originality:** 2
**Rating:** 3
**Confidence:** 4

**Summary:**

This paper is interested in randomly picking a feasible (usually meaning independent set in a matroid) set that maximizes a notion _fairness_ (the minimum probability that an element is fixed) online. More specifically, the model is: elements are arriving one by one, and we must decide whether or to choose the element to be in our output; we would like a scheme which maximizes fairness.

It turns out randomly picking fair independent sets is closely related to $\alpha$, the arboricity of a matroid. There's a natural lower bound of $\Omega(\alpha)$ to the fairness of any scheme. The authors come up with several algorithms; for general matroids, they come up with a $O(\alpha \cdot \log k)$ ($k$ is the rank) scheme and for more specific matroids, they come up with $O(\alpha)$ schemes.

**Questions:**

Just what I have in the "Weaknesses" section of the review.

**Ethical Concerns:**

["NO or VERY MINOR ethics concerns only"]

**Final Justification:**

I still lightly reject this paper right now because 1) the main matroidal theorem proof is not so novel 2) no log n lower bound is presented and 3) no experiments are given to showcase this notion of fairness on benchmarks / examples.

**Limitations:**

Yes.

**Paper Formatting Concerns:**

I didn't notice any major formatting issues.

**Quality:**

2

**Strengths And Weaknesses:**

Weaknesses:

The central result seems to just follow from almost folklore set cover arguments. It's known that you can Greedily, online, color at least half of the elements with $\alpha$ colors (follows from the fact that a maximal common independent set of a matroid intersection has at least half the cardinality of a maximum common independent set). And so that means that it would take at most $\log k$ passes to color all the elements which is equivalent to using at most $\log k \cdot \alpha$ colors. Indeed, this is a lot like the argument the authors seem to write line lines 229 to 240. I am not sure if there is any extra nuance there that I am missing.

What I'm really worried about is the following: can't online contention resolution schemes (https://arxiv.org/abs/1508.00142) get you an $O(\alpha)$ approximation online? A $\gamma$ contention resolution scheme takes in a feasible vector $x$ and outputs an independent set $I$ such the probability that an element $e$ appears in $I$ is at least $\gamma \cdot x_e$. This is a very rich field with a lot of work, even in the online setting. And indeed, there is a constant factor online contention resolution scheme... If I set $x_e = \frac{1}{\alpha}$ for all elements $e$ (which is feasible by the definition of $\alpha$), then I get a $O(\alpha)$ fair algorithm. Granted, I need to know $\alpha$ up front with the scheme I just stated here, but I think we can even get rid of that with a guess and double addition.

If what I'm saying is right, I think it supersedes many of the main results of this paper, and is even tight.

Also, I'm not sure why this is submitted to NeurIPS. It's a wonderful theoretical question and deserves to be studied in the theory community. I'm not sure what the relevance of this paper is to NeurIPS' more practical focus, except for appealing to fairness. I love matroid coloring as a theory problem, but I am not convinced of the practical / experimental merits of this paper. Should the authors be interested in NeurIPS as a venue, more thought and content of the paper should be devoted to applications.

Strengths:

A really really nice question, with beautiful objects (matroids, arboricity, fairness, etc). The paper is well written as well. The authors work on both lower and upper bounds. I like this question.

---

> ### Author Rebuttal · Authors · 2025-07-30
>
> We thank reviewer eEjL for your thorough feedback.
>
> **Connection to OCRS**
>
> Regarding your specific suggestion of setting $x_e = 1/\alpha$ for all $e$, we would like to clarify that in our setting, the arriving matroid is not known in advance. If it were, the problem would reduce to the offline setting analyzed in Theorem 1.
> More broadly, we address the connection to Online Contention Resolution Schemes (OCRS) in Appendix A. While we initially observed a similarity between our formulation and OCRS, key differences prevent us from directly applying techniques from that line of work. Most notably, OCRS assumes that the set of arriving elements is sampled from a known distribution. In our setting, by contrast, the arrival sequence is chosen adversarially, which precludes the use of OCRS tools that rely heavily on preprocessing based on the distribution.For example, the OCRS approach of Feldman et al. [FSZ16] for matroids involves recursively identifying elements that are “in danger” of being spanned by other active elements and contracting them, a process that requires distributional knowledge (see Section 2.1 of their paper). Without access to this distribution, such a decomposition is not possible.
> Additionally, the original OCRS formulation in [FSZ16] assumes that active elements are selected independently. While the recent work of Dughmi [Dug21] extends OCRS to certain correlated settings, these extensions rely on distributional knowledge and are limited to random arrival order. Moreover, [Dug21] does not provide algorithmic results; instead, it shows that in such settings, the problem reduces to the well-known matroid secretary problem, which remains unresolved. In fact, Dughmi proves that no meaningful guarantees are possible under adversarial arrival or when the distribution is unknown. This highlights a key distinction between OCRS and our model: despite the adversarial arrival order and adversarial choice of elements, our approach yields meaningful guarantees for general matroids (see Theorem 2). Finally, we note that the lower bound in [Dug21] (Example 3.14) does not apply to our setting, as it critically depends on making all elements active with small probability. In contrast, our model requires that the set of active elements always satisfy the arboricity constraint of at most $\alpha$.
>
> **Choice of venue**
>
> We believe NeurIPS is an ideal venue for this work, as it addresses an online selection problem motivated by fairness considerations. Online selection problems have been extensively studied in prior NeurIPS and ICML papers [1–6]; notably, [5] focuses specifically on OCRS.
> The matroid constraint is also commonly considered as a solution requirement (e.g., see [1, 7-10]), Given the substantial body of work on online selection, matroids, and fairness in NeurIPS, we view it as a particularly appropriate venue for our contribution.
>
>
> If you feel that our response has adequately addressed your major concerns, we would appreciate it if you could possibly adjust your score accordingly.
>
> **References**:
>
> [1] Antonios Antoniadis, Themis Gouleakis, Pieter Kleer, and Pavel Kolev. Secretary and online matching problems with machine learned advice. In NeurIPS, 2020.
>
> [2] Eric Balkanski, Will Ma, and Andreas Maggiori. Fair secretaries with unfair predictions. In NeurIPS, 2024.
>
> [3] Ziyad Benomar, Dorian Baudry, and Vianney Perchet. Lookback prophet inequalities. In NeurIPS, 2024.
>
> [4] Wei Tang, Haifeng Xu, Ruimin Zhang, and Derek Zhu. Intrinsic robustness of prophet inequality to strategic reward signaling. In NeurIPS, 2024.
>
> [5] Vasilis Livanos. Simple and optimal greedy online contention resolution schemes. In NeurIPS, 2022.
>
> [6] Yuan Deng, Vahab Mirrokni, and Hanrui Zhang. Posted pricing and dynamic prior-independent mechanisms with value maximizers. In NeurIPS, 2022.
>
> [7] Paul Dütting, Federico Fusco, Silvio Lattanzi, Ashkan Norouzi-Fard, and Morteza Zadimoghaddam. Fully dynamic submodular maximization over matroids. In ICML, 2023.
>
> [8] Shengjie Wang, Tianyi Zhou, Chandrashekhar Lavania, and Jeff A. Bilmes. Constrained robust submodular partitioning. In NeurIPS, 2021.
>
> [9] Orestis Papadigenopoulos and Constantine Caramanis. Recurrent submodular welfare and matroid blocking semi-bandits. In NeurIPS, 2021.
>
> [10] Marwa El Halabi, Federico Fusco, Ashkan Norouzi-Fard, Jakab Tardos, and Jakub Tarnawski. Fairness in streaming submodular maximization over a matroid constraint. In ICML, 2023.

---

> > ### Author Response · Authors · 2025-08-03
> >
> > Dear Reviewer,
> >
> > We would like to kindly check whether you have any remaining questions or concerns. We would be happy to provide further clarification. If not, we would be grateful if you could consider updating your score in light of the rebuttal.

---

> > > ### Comment · Reviewer_eEjL · 2025-08-05
> > >
> > > Hello,
> > >
> > > You're right about how OCRS doesn't cover this; they do need to know the whole matroid up front (queries about the spans and what not) in order to make the decomposition. In your setting it seems the matroid is revealed online. I will update my score to be a 3 instead of 2.
> > >
> > > I still lightly reject this paper right now because 1) the main matroidal theorem proof is not so novel 2) no log n lower bound is presented and 3) no experiments are given to showcase this notion of fairness on benchmarks / examples.

---

### Official Review · Reviewer_nMZ2 · 2025-07-02

**Clarity:** 3
**Significance:** 3
**Originality:** 3
**Rating:** 5
**Confidence:** 2

**Summary:**

In the online matroid problem, elements of an unknown matroid arrive online over time. The goal is to select an independent set of this unknown matroid. To this end, an algorithm has to irrevocably decide upon arrival of each element whether to include the element or not. The authors study this problem from a fairness perspective, where the objective is to maximize the minimum selection probability over all elements.

First, the authors show that the offline optimal solution value is in the interval $[\alpha-1,\alpha]$ by exploiting a classical result by Edmonds, where $\alpha$ is the arboricity of the matroid which is the minimum number of independent sets necessary to partition the ground set of elements. As a first algorithmic main result, the authors show that there is a $\tilde{\mathcal{O}}(\alpha \log k)$ fair algorithm for general matroids and adversarial arrivals, where $k$ is the rank of the matroid. The algorithm is based on an  online matroid coloring inspired greedy algorithm for a relaxed setting, where $\alpha$ and the rank $k$ are known to the algorithm. To generalize to the fully oblivious setting, the authors give a blackbox reduction that generalizes algorithms for the relaxed setting to the fully oblivious one, with a logarithmic  loss. Following a similar framework but exploiting additional structural properties, they give an improved result for the special case of laminar matroids.

Afterwards, the authors move on to the random order arrival setting. They show that this setting can be reduced to a different, sample-based setting. In this setting, several matroid parameters can be efficiently approximated by exploiting the possibility to sample, which allows a reduction to the partially oblivious setting without logarithmic loss and immediately allows improved algorithms for general matroids by using the adversarial result with a different reduction. As a final main result, the authors prove an improved result for graphic matroids.

**Questions:**

* Has your objective function, maximizing the minimum selection probability, been studied for different problems?

**Ethical Concerns:**

["NO or VERY MINOR ethics concerns only"]

**Final Justification:**

As outlined in my original review, I think that the paper is a good contribution. In my opinion, the study of matroid selection with a fairness objective is a relevant and interesting question. On the technical side, I think that the presented proofs and algorithms are a nice combination of classical results and new ideas. The presented blackbox reductions might be particularly useful for future works. My concern regarding the missing discussion of previous works with similar objectives has been addressed in the rebuttal. Therefore, I would like to keep my positive score. I selected a low confidence of 2, because I am not extremely familiar with previous related works. Therefore, my estimation of the technical contribution should be treated with some scepticism.

**Limitations:**

yes

**Quality:**

3

**Strengths And Weaknesses:**

Strength:
* In my opinion, the paper is very well-written. The main part does a very good job of highlighting the key ideas behind the algorithms and proofs. The general problem as well as the considered special cases are well motivated. I enjoyed reading the paper.
* Multiple results presented in this paper feature a nice combination of classical results, (such as Edmonds result on the arboricity, known insights on online matroid coloring and further structural results on laminar and graphic matroids) with new ideas specific to the problem at hand.
* The paper contains two blackbox reductions that might be particularly useful for future works: A reduction from the totally oblivious model to the partially oblivious model, and a reduction from the random order setting to a random sampling setting.

Weaknesses:
* The oblivious online setting should be clarified a bit. The paper seems to assume access to an oracle that can check independence for the already revealed elements. This should explicitly be stated somewhere.
* Maybe I overlooked something, but the related work section does not seem to discuss previous works that consider the objective of maximizing the minimum selection probability. Is this the first work to consider such a fairness measure or is the discussion of previous works just missing? In the former case, this should be explicitly stated. In the latter case, adding such a discussion would be helpful.
* This is very minor, but technically you probably have to exclude loops in order for the formula of $\alpha$ in Theorem 6 to be well-defined.

---

> ### Author Rebuttal · Authors · 2025-07-30
>
> We thank reviewer nMZ2 for your thorough feedback and positive review.
>
> > Has your objective function, maximizing the minimum selection probability, been studied for different problems?
>
> Regarding similar notions in prior work, the objective of maximizing the
> minimum selection probability has indeed been considered in prior work. For
> instance, Flanigan et al. [1] study the problem of selecting citizens’
> assemblies and propose algorithms that aim to maximize minimum selection
> probabilities. Several works also consider fairness objectives that are
> conceptually similar in the context of influence maximization [2, 3, 4] and the Santa Claus problem which we discuss in the paper. However, to the best of our knowledge, we are the first to consider such fairness objectives in a secretary-like setting where elements arrive online
> and selections must satisfy a combinatorial constraint. This setting introduces distinct algorithmic challenges that require different techniques from those explored in prior work. We will make sure to revise our submission in order to include this discussion.
>
>
>
>
> > The oblivious online setting should be clarified a bit…
> > This is very minor, but technically you probably have to excluded loops…
>
> We acknowledge the point about explicitly defining the model to include access to an independence oracle, and we will make sure to clarify this in future revisions of the paper. We also acknowledge the minor point about loops in the definition of arboricity and will include that in the revisions as well.
>
> References:
>
> [1] Bailey Flanigan, Paul Gölz, Anupam Gupta, Brett Hennig, and Ariel D. Procaccia. Fair algorithms for selecting citizens’ assemblies. In Nature, 2021.
>
> [2] Ruben Becker, Gianlorenzo D’Angelo, Sajjad Ghobadi, and Hugo Gilbert. Fairness in influence maximization through randomization. In Journal of Artificial Intelligence Research, 2022.
>
> [3] Alan Tsang, Bryan Wilder, Eric Rice, Milind Tambe, and Yair Zick. Group-fairness in influence maximization. In IJCAI, 2019.
>
> [4] Benjamin Fish, Ashkan Bashardoust, Danah Boyd, Sorelle Friedler, Carlos Scheidegger, and Suresh Venkatasubramanian. Gaps in information access in social networks?. In The Web Conference (WWW), 2019.

---

> > ### Comment · Reviewer_nMZ2 · 2025-08-04
> >
> > Thank you to the authors for their response, especially for providing additional references on the objective function.

---

### Official Review · Reviewer_ixyh · 2025-07-03

**Clarity:** 3
**Significance:** 4
**Originality:** 4
**Rating:** 4
**Confidence:** 4

**Summary:**

This paper studies the problem of sequentially selecting elements of a matroid in an online manner while ensuring that each element has a guaranteed minimum probability of selection, introducing the notion of *f*-fairness as a principled fairness criterion that contrasts with classical objectives focused solely on maximizing total utility.


They show that the optimal fairness guarantee is $[\alpha - 1, \alpha]$, where $\alpha$ is the arboricity of the matroid. If the matroid is known in advance (the offline case), it is straightforward to design an algorithm that achieves $\alpha$-fairness.

For the online problem, they consider adversarial order and random order models. In the adversarial model, they show that for general matroids it is possible to achieve an $\alpha(\ln k + 1)$-fair algorithm, where $k$ is the rank. While their algorithm requires knowing $\alpha$ and $k$ in advance, they show that an algorithm can be made totally oblivious—i.e., without any prior knowledge—by incurring only an additional polylogarithmic factor loss in fairness.

They are able to improve the guarantees for laminar matroids, where laminar matroids are defined by a hierarchical family of subsets with associated capacity constraints. They achieve a $(2\alpha - 1)$-fair algorithm by designing an approach that maintains an online ordering of elements and applies a careful coloring scheme to ensure feasibility while preserving fairness.

Next, they study random order fair matroid selection, and their main result is a $(4 + o(1))\alpha$-fair algorithm for graphic matroids. This algorithm learns an approximate degeneracy ordering from a sample of edges and then uses it to guide the selection process, achieving fairness guarantees that are nearly tight up to constant factors.

**Questions:**

Please give more intuition about why it is not possible to achieve $O(\alpha)$-fairness for all matroids in the random order or adversarial settings.

**Ethical Concerns:**

["NO or VERY MINOR ethics concerns only"]

**Final Justification:**

The authors’ response addressed several of my concerns, and I considered increasing the score. In the end, I chose to maintain the current score.

**Limitations:**

Yes.

**Paper Formatting Concerns:**

No.

**Quality:**

3

**Strengths And Weaknesses:**

Overall, I find their model very interesting, and their results are both clear and well-motivated. While they do not introduce fundamentally new techniques, the paper applies existing tools—such as online coloring schemes and degeneracy orderings—in a thoughtful and effective way to establish strong fairness guarantees. The presentation is rigorous, and the contributions represent a valuable addition to the literature on online selection and fairness. I am inclined to recommend acceptance.

---

> ### Author Rebuttal · Authors · 2025-07-30
>
> We thank reviewer ixyh for your thorough feedback. In response to your question, we first note that we do not prove that it is not possible to achieve $O(\alpha)$-fairness for all matroids in the random order or adversarial settings (specifically, we do not prove this in the setting where knowledge of $\alpha$ is assumed; we do prove it in the setting where such knowledge is not assumed, and in fact even prove it for rank 1 matroids in that setting, with the intuition there being that early elements must be accepted with high probability, and the required probability decays slowly enough that the total probability of acceptance must diverge).
>
> However, we can offer some intuition for why it may be difficult to achieve $O(\alpha)$-fairness not only for general matroids, but even for graphical matroids, in the adversarial order model. Consider a more restrictive version of the adversarial order setting for graphical matroids where when accepting an edge, we not only must satisfy independence constraints, but we also must match said edge to one of its endpoints, such that for each vertex v we match at most one edge to v. Thus, we in effect partition the graphical matroid by vertex. In this setting, it is not possible to do better than $O(\alpha \cdot k)$-fairness (which is tight). We provide a sketch of the proof of this lower bound at the end of our response.
>
> This lower bound is notable because our algorithm for graphical matroids in the random order setting obeys this constraint: it groups edges by node and selects one edge for each node. Furthermore, this is a key property of algorithms for the related matroid secretary problem, as we note in our paper on lines 334-342. Thus, the fact that grouping edges by vertex in this way cannot give us anything better than $O(\alpha \cdot k)$-fairness suggests that breaking this $\lg k$ boundary is difficult, requiring new techniques that do not tie an edge to a specific endpoint, and perhaps impossible (i.e. the algorithm for general matroids is tight).
>
> If you feel that our response has adequately addressed your major concerns, we would appreciate it if you could possibly adjust your score accordingly.
>
> **Proof sketch for the lower bound we described above**:
>
> Let $k$ be such that $k + 1$ is a power of $2$; we proceed in $\lg (k + 1)$ iterations. In iteration $j$ (numbered starting from $1$), we present to the algorithm $2^{\lg (k + 1) - j}$ edges such that no two of these edge share a common endpoint. In the first iteration these edges (of which there are $\frac k 2$) can be chosen arbitrarily. In each later iteration $j > 1$, we choose the $2^{\lg (k + 1) - j}$ edges by:
>
> 1. Pairing up the edges in the previous iteration arbitrarily.
> 2. For each pair $(e_1, e_2)$, selecting an endpoint $u$ uniformly at random from the endpoints of $e_1$ and an endpoint $v$ uniformly at random from the endpoints of $e_2$, and choosing the edge $(u, v)$.
>
> In this way, by the end we will have presented not only a graph, but in fact a tree with $k$ edges (and thus with rank $k$). As a tree, the graph has arboricity $1$. We now show how an algorithm dealing with this distribution cannot be better than $\frac {\lg (k + 1)} 2$-fair.
>
> Consider an algorithm $A$ that is $\frac 1 p$-fair, meaning that it accepts each edge with probability at least $p$. For each edge $e$ that we present, define $q_e$ to be the expected number of endpoints of $e$ which have been matched by the end of iteration $j$, where $j$ is the iteration in which $e$ is presented.
>
> We can show inductively that for any $j$, for all edges $e$ presented in iteration $j$, we have $q_e \geq jp$. For brevity, consider there to be an iteration $0$ for which the claim is obvious. Now, given that the claim holds for iteration $j - 1$, first consider any edge $e$. $e$ is defined from a pair $e_1, e_2$ of edges in the previous iteration, which therefore satisfy $q_{e_1}, q_{e_2} \geq (j - 1)p$. Then, $u$ is uniformly randomly selected from the endpoints of $e_1$. We then have that the expected probability of $u$ having been matched by the end of iteration $j - 1$ is at least $\frac {(j - 1)p} 2$. The same then holds for $v$. Thus, the expected number of endpoints of $e$ which have been matched by the end of iteration $j - 1$ is $(j - 1)p$. In iteration $j$, we then match $e$ itself to one of its endpoints with probability at least $p$. It thus follows that $q_e \geq (j - 1)p + p = jp$.
>
> Given this claim, we now consider the single edge $e$ added in the final iteration. As the final iteration is iteration $\lg(k + 1)$, we must have that $q_e \geq \lg(k + 1)p$. However, note that as $q_e$ is the expected number of endpoints of $e$ that are matched by the end of the iteration, and $e$ has two endpoints each of which can be matched at most once, it must be that $q_e \leq 2$. Therefore, $\lg(k + 1)p \leq 2$, and so $p \leq \frac 2 {\lg(k + 1)}$. Thus, $A$ can be at best $\frac {\lg(k + 1)} 2$-fair, completing the proof.

---

> > ### Author Response · Authors · 2025-08-03
> >
> > Dear Reviewer,
> >
> > We would like to kindly check whether you have any remaining questions or concerns. We would be happy to provide further clarification. If not, we would be grateful if you could consider updating your score in light of the rebuttal.

---

> > > ### Comment · Reviewer_ixyh · 2025-08-06
> > > **Further clarfication**
> > >
> > > Thank you for the detailed response. Regarding the lower bound, you proved it to a more restricted setting and wrote: “Thus, the fact that grouping edges by vertex in this way cannot yield anything better than $O(\alpha k)$-fairness suggests that breaking the $O(\alpha \log k)$ barrier is difficult.” Should this be understood as a formal statement, or merely as intuition?

---

> > > > ### Author Response · Authors · 2025-08-07
> > > >
> > > > Thank you for your followup.
> > > > The result can be formally shown in the restricted setting; the reason we discuss this restricted setting however is to give an intuition of why we believe the problem is difficult in the general setting. Thus, the sentence you quote is indeed intuition.

---

> > > > > ### Comment · Reviewer_ixyh · 2025-08-08
> > > > >
> > > > > Thank you for your response. I have decided to maintain my slightly positive score.

---

### Decision · Program_Chairs · 2025-09-17

**Decision:**

Accept (poster)

**Comment:**

This paper introduces the $f$-fair matroid selection problem. Elements from an unknown matroid arrive online sequentially, and the goal is to choose elements, such that the chosen elements form an independent set, and for any element $e$, the probability of choosing $e$ is at least $1/f$. The authors first study the offline optimal solution, and then give algorithmic results for the online setting with only a logarithmic factor overhead over the offline solution. They also obtain improved results for the families of laminar and graphic matroids, and they additionally study the random arrival model.

The reviewers were in agreement about the originality of their model and the elegance of the ideas involved. They also were unanimous in praising the clear exposition.

A few reviewers were concerned that the new algorithmic techniques were somewhat incremental, but it's clear that the paper's originality lies in tying together existing techniques to solve a novel, relevant, well-posed problem. In the discussion phase, the authors mostly successfully responded to other issues that were raised. We'd encourage the authors to incorporate this discussion into the final revision.